# Decoupling agriculture pollution and carbon reduction from economic growth in the Yangtze River Delta, China

**Ruxue Yuan**[1,2,3], **Caiyao Xu**[1,2,3]*, **Fanbin Kong**[1,2,3]*

**1** Institute of Ecological Civilization, Zhejiang A&F University, Zhejiang, China, **2** Research Academy for Rural Revitalization of Zhejiang Province, Zhejiang A&F University, Zhejiang, China, **3** College of Economics and Management, Zhejiang A&F University, Zhejiang, China

* kongfanbin@aliyun.com (FK); xucaiyao@zafu.edu.cn (CX)

## Abstract

Agriculture is the foundation of the national economy, and agricultural nonpoint source pollution and carbon emissions are the main environmental problems limiting the development of the agricultural economy. This study takes the Yangtze River Delta as the research object and measures agricultural carbon emissions and nonpoint source pollution in the study area from 2010 to 2020 respectively. The Tapio decoupling model is used to study types of decoupling between agricultural pollution and carbon reduction and economic growth in the Yangtze River Delta from 2010 to 2020, and the GM (1,1) model is used to predict the decoupling relationship between the agricultural environment and economic growth over the next ten years. The results show the following: (1) Agricultural COD emissions come mainly from livestock and poultry breeding, dropped from 1,130,120 tons in 2010 to 908,460 tons in 2020. Agricultural TN and TP emissions come mainly from plantations. Agricultural TN emissions dropped from 892,310 tons in 2010 to 788,020 tons in 2020. Agricultural TP emissions dropped from 149,590 tons in 2010 to130,770 tons in 2020. Agricultural carbon emissions dropped from 17,115,900 tons in 2010 to 15,786,600 tons in 2020, and come mainly from agricultural fertilizer and diesel fuel and pig breeding. (2) The decoupling effect of agricultural pollution reduction and carbon reduction in the Yangtze River Delta and economic growth has been in a long-term state, with negative decoupling occurring in a few regions, mainly in 2011, 2014 and 2020. (3) In the next ten years, except for 2021, when the coordination between agricultural pollution reduction and economic growth is poor, the two show good decoupling in the remaining years. Based on the results, this study makes recommendations on how to carry out comprehensive environmental management and promote green agricultural development.

## Introduction

Since the reform and opening up, China's agricultural production conditions have continued to improve, providing strong support for the development of agricultural modernization, with an average annual growth rate of 4.6% in the gross agricultural product (AGDP) from 1978 to

**Data Availability Statement:** The data on gross domestic product, rural population, fertilizer application, year-end stock of livestock and poultry breeding and fish production are from the 2011-

2021 Shanghai Statistical Yearbook, Jiangsu Statistical Yearbook, Zhejiang Statistical Yearbook and Anhui Statistical Yearbook, as well as statistics from cities and counties. The coefficients for rural domestic production and emission were obtained from the "Coefficients and Instructions for the Use of Production and Emission of Domestic Sources" (2011 revised edition), the coefficients for pig, cattle and aquatic products production and emission were obtained from the "Handbook on Accounting Methods and Coefficients for Agricultural Sources Production and Emission" (2021 edition).

**Funding:** This research was supported by the Zhejiang Provincial Natural Science Foundation of China (Grant No. Z22D010686), the National Natural Science Foundation of China (Grant No. 42071283), the Zhejiang Soft Science Research Program of China (Grant No. 2022C35104), the Research Development Fund of Zhejiang A&F University (Grant No. 2020FR066), and the Special Project of Cultivating Leading Talents in Philosophy and Social Science of Zhejiang Province (Grant No. 21YJRC12-2YB and 21YJRC2ZD)". The funders had no role in study design, data collection and analysis, decision to publish, or preparation of the manuscript.

**Competing interests:** The authors have declared that no competing interests exist.

2020 [1]. However, the development of the agricultural economy has generated agricultural pollution, and with the disappearance of the demographic dividend, environmental pollution and resource depletion have become increasingly prominent and counterproductive constraints on the development of the agricultural economy [2, 3]. The current situation of agricultural nonpoint source pollution in China is very serious [4], and the total amount and intensity of carbon emissions from agricultural land use are increasing [5]. The COD, TN, and TP pollution loads from Chinese agriculture increased by 91.0%, 196.2%, and 244.1% from 1978 to 2017 [6], the average annual growth rate of $CO_2$ emissions is 0.78% lower than the average annual growth rate of agricultural output of 3.82% from 1997 to 2014 [7], the conflict between agricultural production and environmental pollution in China and each region has seriously deteriorated since 2012 [8]. Agricultural production is highly dependent on the environment and climate, and accurately characterizing the relationship between environmental pollution in agriculture and economic growth is an important topic that needs attention. Therefore, it is important to study the decoupling of agricultural pollution and carbon reduction from economic growth to achieve sustainable agricultural development. According to data from the Second National Pollution Census of China and the Food and Agriculture Organization of the United Nations, China's agricultural chemical oxygen demand (COD), total nitrogen (TN), and total phosphorus (TP) emissions reached 10,671,300 tons, 1,414,900 tons, and 212,000 tons, respectively, in 2017, accounting for half of all pollution emissions. Global greenhouse gas emissions from agricultural land exceeded 30% of total global anthropogenic greenhouse gas emissions, and China's agricultural source greenhouse gas emissions accounted for approximately 17% of total national greenhouse gas emissions [9]. In addition, agriculture is not only a "contributor" to carbon emissions, but also one of the sectors most vulnerable to climate change. As the global warming trend intensifies, agricultural production will become limited; thus, it is crucial to balance the relationship between economic development and agricultural carbon emissions and agricultural nonpoint source pollution. China proposed a "double carbon" goal at the 75th United Nations General Assembly and incorporated it into its ecological civilization system in 2020. In December 2020, the Central Economic Work Conference proposed that "we should continue to fight the battle against pollution and achieve the synergistic effect of pollution reduction and carbon reduction" and included this goal in the 14th Five-Year Plan for national economic and social development. The No. 1 Document of the Central Government in 2022 proposed strengthening the comprehensive management of agricultural nonpoint source pollution and promoting green development in agriculture and rural areas. Fertilizers, pesticides and agricultural films are the main sources of not only agricultural nonpoint source pollution but also greenhouse gas nitrous oxide emissions, so reducing the consumption of fertilizers, pesticides and agricultural films is a synergistic strategy to reduce pollution and carbon emissions [10]. Therefore, we can start with agricultural production behavior to promote the decoupling of pollution and carbon reduction from economic growth and help achieve sustainable agricultural development.

According to calculations based on data published in the 2020 China Environmental Statistics Yearbook, in 2019, the amount of agricultural fertilizer applied in the Yangtze River Delta region reached 6,642,000 tons, the number of pesticides used reached 197,010 tons, the amount of plastic film used was 297,840 tons, and the amount of agricultural COD and agricultural ammonia nitrogen in wastewater discharge accounted for 8.856% and 22.672% of the country's total, respectively, posing a serious threat to the ecology of China's watersheds. Therefore, this study takes 41 municipalities in the Yangtze River Delta as the research object, uses the Tapio model to analyze the decoupling relationship between pollution reduction and carbon reduction and economic growth in the Yangtze River Delta from 2011 to 2020. It predicts the future decoupling state between the two variables from 2021 to 2030 by using a gray

prediction model. The remainder of the article is organized as follows. Section II compares the literature on the subject in terms of agricultural nonpoint source pollution and agricultural carbon emissions. Section III describes the study area overview, research methodology and data sources. Section IV presents the study results. Section V reports and analyzes the study results and gives policy recommendations, Section VI presented the conclusions and limitations of the study.

## Literature review

Agricultural nonpoint source pollution is an important factor in the deterioration of air and water quality. The vast size of China and the many types of agricultural production operations make it difficult to identify, monitor, and prevent and control the sources of agricultural nonpoint source pollution emissions [11]. Due to the effective implementation of government policies, digital modeling and the application and development of remote sensing technology, significant progress has been made in solving the problem of agricultural nonpoint source pollution [12]. Studies on agricultural nonpoint source pollution at home and abroad have focused on pollutant monitoring and load calculation [13–16]; pollutant accounting and distribution characteristics [17]; and pollution prevention and control, emission reduction and impact factors [18, 19]. Studies have shown that China's Taihu Lake Basin suffers the most serious agricultural nonpoint source pollution [20], while Chaohu Lake Basin shows an overall increasing trend of TN pollution and a small increasing trend of TP pollution [21]. The environmental behavior of farmers and the landscape pattern of the basin influence the emission and distribution of agricultural nonpoint source pollution from both anthropogenic and natural aspects [22–24]. The systematic "4R" approach of source reduction, process retention, nutrient reuse and water restoration is an effective measure to reduce nitrogen and phosphorus loads on farmland and improve the ecology of water bodies in the lower Yangtze River [25].

The global warming phenomenon has seriously affected human production and life, and greenhouse gas emissions are an important factor in global warming. As the global warming trend gradually intensifies, scholars from various countries are exploring ways to reduce greenhouse gas emissions, mainly by reducing carbon emissions. Domestic and international studies on agricultural carbon emissions have focused on the distribution of agricultural carbon sources/sinks and carbon footprints [26–28]; the impact mechanism of agricultural carbon emissions [29, 30]; the mechanism of agricultural carbon emission reduction and its cost-benefit analysis [31–33]; and the relationship between carbon emissions and economic development [34, 35]. Studies have shown that agricultural growth in developing countries has a positive impact on $CO_2$ emissions, and with economic growth, $CO_2$ emissions have not yet reached the turning point of the environmental Kuznets curve [36]; there is a two-way causal relationship between agricultural carbon emissions and agricultural economic growth in both the short and long term in China's major grain-producing regions [37]; China's agricultural carbon emissions show an inverted trend as "the overall growth rate of carbon emissions has shown a gradual decline" [38]. In terms of mitigating agricultural carbon emissions, the use of renewable energy sources such as solar energy and biomass can effectively reduce agricultural carbon emissions and help achieve carbon neutrality [39, 40]. Although a more unified methodological system has been developed for the accounting of agricultural nonpoint source pollution and agricultural carbon emissions, consistent results have not yet emerged due to different data sources and specific categories of accounting, and some results differ greatly.

The Central Economic Work Conference in December 2020 proposed that "we should continue to fight the battle against pollution and achieve synergistic effects of pollution reduction and carbon reduction". Studies on carbon reduction are still at the level of legal system, theoretical level and realization path, but there is a lack of corresponding empirical research and

quantitative analysis. At the theoretical level, Zheng et al. [41] and Fei et al. [42] have explored paths to reducing pollution and carbon in relation to the "double carbon" target. At the empirical level, Wang et al. [43] constructed a comprehensive pollution reduction-carbon reduction-economy evaluation index system to study differences between pollution reduction and regional economic development levels in China and showed that the ternary coupling of pollution reduction-carbon reduction-economy in the southeastern coastal region was better.

The relationship between the economy and the environment has been extensively researched by scholars at home and abroad, especially in developing countries that are facing the pressures of industrialization and urbanization [44]. Studies on the environment and the economy have focused on validating the environmental Kuznets curve, measuring the degree of decoupling between the two and measuring the coordination degree of ecology and economy [45]. Among these approaches, the Tapio model is considered to be a more effective tool for conducting decoupling analysis [46], and previous studies have analyzed the factors of decoupling. Studies of the drivers of decoupling mostly use the log mean divisor index (LMDI) method. China has shown weak decoupling between economic development and carbon emissions over the years [47], strong decoupling with PM2.5 emissions [48], and a decoupling index with energy consumption that fluctuates within a relative decoupling band [49]. Measures such as increased investment in research and development and the use of renewable energy can promote decoupling; however, factors such as urbanization inhibit decoupling [50, 51]. Notably, the current decoupling theory is applied mostly to secondary and tertiary industries, such as construction [52], transportation [53] and industrial production [54]. The application of decoupling theory in the agricultural sector is relatively rare, especially in the study of the relationship between agricultural nonpoint source pollution and the agricultural economy. In addition, most studies take the provincial area as the basic unit or a specific city as the research object, lacking the concept of region. Therefore, this paper starts from the agricultural field and takes prefecture-level cities as the research object to explore the relationship between agricultural pollution reduction and carbon reduction and economic growth in the Yangtze River Delta region of China to provide theoretical support for realizing sustainable agricultural development in the context of "double carbon".

Given this goal, this paper makes the following three contributions. (1) At the research level, previous studies have taken the provincial-level unit or an individual city as the object of study. This paper draws on the previous research methods and takes 41 prefecture-level cities in the Yangtze River Delta as the object of study, and makes a longitudinal and cross-sectional comparison of each region, which makes up for the deficiencies of previous studies from the spatial perspective. (2) In terms of research content, previous studies have only explored the relationship between agricultural pollution and economy. This study examines the relationship between non-point source pollution from agriculture and carbon emission from the concept of pollution reduction and carbon reduction, providing ideas and scientific basis for reducing three-dimensional pollution at source and achieving synergistic management of agricultural environment and climate. (3) This study not only investigates the decoupling between agricultural pollution and economy in the Yangtze River Delta, but also predicts the decoupling trend between agricultural pollution and carbon reduction and ecological and economic growth in the next decade, which is extended in the time range.

## Materials and methods

### Study area

The Yangtze River Delta (hereafter "the YRD") is located in the lower reaches of the Yangtze River, an alluvial plain formed before the Yangtze River enters the sea, and covers 41 cities in

Jiangsu, Zhejiang, Shanghai and Anhui, with an area of 358,000 square kilometers. First, the YRD is an important production area for commercial foodstuffs and a pilot demonstration area for sustainable agricultural development, which is strategically important in promoting national agricultural development and maintaining food security. Second, the spread of agricultural pollutants will lead to serious soil and water pollution. The YRD is bordered by the Yellow Sea and the East China Sea, with a dense network of rivers and streams, and offers a geographical advantage for studying environmental pollution in agriculture. Finally, according to calculations based on data published in the 2020 China Environmental Statistics Yearbook, in 2019, the amount of agricultural fertilizer applied in the YRD region reached 6,642,000 tons, pesticide use reached 197,010 tons, the amount of plastic film used was 297,840 tons, and the amount of agricultural COD and agricultural ammonia nitrogen in wastewater discharge accounted for 8.856% and 22.672% of the country's total, respectively. These conditions pose a serious threat to the ecology and health of China's watersheds; thus, it is of great practical importance to use the YRD as the study area.

## Data sources

Data on the gross domestic product, rural population, fertilizer application, year-end stock of livestock and poultry breeding and fish production were obtained from the 2011–2021 Shanghai Statistical Yearbook (sh.gov.cn), Jiangsu Statistical Yearbook (jiangsu.gov.cn), Zhejiang Statistical Yearbook (zj.gov.cn) and Anhui Statistical Yearbook (ah.gov.cn), as well as from cities and counties. The coefficients for pig, cattle and aquatic product production and emission were obtained from the Handbook on Accounting Methods and Coefficients for Agricultural Sources Production and Emission (2021 edition) pdf (mee.gov.cn). As the handbook did not provide coefficients for sheep and poultry production and emission, the coefficients for agricultural fertilizers, sheep and poultry production and emission were drawn from Guo et al. [17], and the carbon emission coefficient data were obtained from Tian et al. [55] and Xia et al. [56].

## Methodology

**Agricultural nonpoint source pollution measurement model.** Agricultural nonpoint source pollution is composed mainly of rural production and livelihoods, where rural production includes farming, livestock farming and aquaculture, with rural farming polluted mainly by agricultural fertilizers and rural daily farming organisms consisting mainly of pigs, cattle, sheep and poultry [57].

$$E = \sum Q_i C_i (EU_i, S) \tag{1}$$

In Eq (1), E is the emission of agricultural and rural pollution; $Q_i$ is the agricultural and rural pollution inventory statistics; $C_i$ is the production and discharge coefficient of pollutant j in unit i, which is determined by unit i and spatial characteristics S, a characterization of the combined impact of regional environment, hydrology and various management techniques on agricultural pollution; the production coefficient is the number of pollutants directly discharged into the environment per unit of pollution inventory; and the discharge coefficient is the amount of pollution released into the environment after treatment. When pollutants are discharged directly, the pollution production factor is equal to the discharge factor. The specific coefficients are shown in **Table 1**.

## Agricultural carbon emission estimation model

According to previous research and the IPCC greenhouse gas inventory, plantation carbon emissions come from six main sources: fertilizers, pesticides, films, machinery, agricultural

**Table 1. Emission factors for each source of agricultural nonpoint source pollution in the YRD.**

| Pollution sources | Pollution inventory | | Production and discharge factors | | | Unit |
|---|---|---|---|---|---|---|
| | | | COD | TN | TP | |
| Crop Planting | Agricultural fertilizers | | 0.000 | 110 | 17.600 | kg/t |
| Livestock Farming | Pig | Shanghai | 6.264 | 0.3461 | 0.0923 | kg/head· year |
| | | Jiangsu | 6.874 | 0.372 | 0.106 | |
| | | Zhejiang | 5.718 | 0.330 | 0.096 | |
| | | Anhui | 5.889 | 0.339 | 0.090 | |
| | Cow | Shanghai | 164.000 | 5.500 | 0.713 | kg/head· year |
| | | Jiangsu | 169.618 | 5.684 | 0.780 | |
| | | Zhejiang | 165.114 | 5.650 | 0.750 | |
| | | Anhui | 164.014 | 5.541 | 0.744 | |
| | Sheep | | 0.242 | 0.121 | 0.020 | kg/head· year |
| | Poultry | | 0.101 | 0.024 | 0.010 | kg/head· year |
| Aquaculture | Aquatic products | Shanghai | 34.446 | 2.299 | 0.358 | kg/t |
| | | Jiangsu | 39.381 | 1.956 | 0.315 | kg/t |
| | | Zhejiang | 20.315 | 2.666 | 0.461 | kg/t |
| | | Anhui | 19.097 | 2.394 | 0.295 | kg/t |

land tilling and agricultural irrigation. According to the IPCC Fourth Assessment Report, the actual calculation converts carbon emission metrics to a uniform standard C according to the conversion standards of 1 kg $CH_4$ = 6.8182 kg C and 1 kg $N_2O$ = 81.2727 kg C [57]. The formula for estimating agricultural carbon emissions is as follows [8].

$$C = \sum C_i = \sum S_i f_i \tag{2}$$

In Eq (2), C denotes total carbon emissions from agriculture, i denotes carbon emission sources, $C_i$ denotes emissions from source i, $S_i$ denotes the amount of carbon emission sources and $f_i$ denotes the carbon emission factor of carbon emission sources. The specific carbon emission factors and data sources are shown in **Table 2**.

## Tapio decoupling model

The decoupling theory is used to describe blocking the link between economic growth and resource consumption or environmental pollution, and pollution decoupling is a process in which the relationship between economic growth and environmental pollution continues to

**Table 2. Emission factors for the main sources of carbon emissions from agriculture in the YRD.**

| Emission sources | | Emission factors | | Unit | Source |
|---|---|---|---|---|---|
| Plantation | Agricultural fertilizers | 0.8965 | | kgC/kg | ORNL |
| | Pesticide use | 4.9341 | | kgC/kg | ORNL |
| | Agricultural plastic film | 5.1800 | | kgC/kg | IREEA |
| | Agricultural diesel | 0.5927 | | kgC/kg | IPCC (2006) |
| | Farmland tilling | 3.1260 | | kgC/hm$^2$ | IABCAU |
| | Agricultural irrigation | 266.4800 | | kgC/hm$^2$ | Tian et al [54] |
| Aquaculture | Category | Enteric fermentation carbon excretion factor | Manure management carbon emission factors | | |
| | Pig | 6.8182 | 27.2728 | kgC/head·year | IPCC (2006) |
| | Cow | 375.0010 | 13.6364 | kgC/head·year | IPCC (2006) |
| | Sheep | 34.0910 | 1.1591 | kgC/head·year | IPCC (2006) |

weaken or even disappear [58]. Tapio first proposed decoupling elasticity in 2005 and classified the types of decoupling into eight categories. In this paper, the decoupling coefficients of agricultural carbon, COD, TN, TP and agricultural economic development of the YRD are measured according to Tapio decoupling model, and the specific calculation formula is shown in Eq (3).

$$a_t = \left( \Delta AP_t \big/ AP_t \right) \Big/ \left( \Delta AGDP_t \big/ AGDP_t \right) \tag{3}$$

In Eq (3), $a_t$ is the decoupling elasticity coefficient for period t, $\Delta AP_t$ and $\Delta AGDP_t$ are the incremental pollutant emissions in period t and the incremental GDP of the primary sector, $\Delta AP_t = AP_t - AP_{t-1}$, $\Delta AGDP_t = AGDP_t - AGDP_{t-1}$. t is the current period and t-1 is the base period.

To better reflect the relationship between carbon emissions and agricultural nonpoint source pollution on agricultural economic development, this study assigns values to the decoupling types, as shown in **Table 3**. The larger the value is, the better the decoupling effect is and the better the coordination between agricultural carbon emissions, nonpoint source pollution and the agricultural economy is.

## Synergistic effect of pollution reduction and carbon reduction

The article uses the synergistic effect coefficient to quantitatively describe the synergistic relationship between agricultural nonpoint source pollution and agricultural carbon emissions, which is calculated as follows.

$$S = \frac{\Delta C / C}{\Delta E / E} \tag{4}$$

In Eq (4), S is the coefficient of the synergistic effect of pollution reduction and carbon reduction, $\Delta C$ is the agricultural carbon emission reduction, C is the agricultural carbon emission, $\Delta E$ is the agricultural nonpoint source pollution emission reduction, and E is the agricultural nonpoint source pollution emission, which is calculated using agricultural COD, TN, and TP, respectively in this paper. $S \leq 0$ indicates that pollution reduction and carbon reduction are not coordinated, and $0 < S < 1$ indicates that the reduction effect of current agricultural activities on agricultural nonpoint source pollution is greater than when S = 1, which indicates that the two effects are comparable. When $S > 1$, current agricultural activities have a greater effect on the reduction in agricultural carbon emissions than on the reduction in agricultural nonpoint source pollution.

**Table 3. Classification of the types of decoupling between agricultural pollution and carbon reduction and economic development.**

| Decoupling status | Variable properties | | The range of values of a | Type of decoupling | Assignment |
|---|---|---|---|---|---|
| | ΔAGDP | ΔAP | | | |
| Decoupling | + | - | a<0 | Strong uncoupling | 8 |
| | + | + | 0≤a<0.8 | Weak decoupling | 7 |
| | - | - | a>1.2 | Decline decoupling | 6 |
| Linking | + | + | 0.8≤a<1.2 | Growth Links | 5 |
| | - | - | 0.8≤a<1.2 | Recession Link | 4 |
| Negative decoupling | + | + | a>1.2 | Expansion negative decoupling | 3 |
| | - | - | 0≤a<0.8 | Weak negative decoupling | 2 |
| | - | - | a<0 | Strong negative decoupling | 1 |

## Gray prediction model

The gray prediction model GM (1.1) is a quantity size prediction model based on historical time-series data, which is suitable for small sample data. This paper predicts the relationship between agricultural pollution and carbon reduction and economic growth in the YRD for the next 10 years based on available agricultural economic data, agricultural carbon emissions and nonpoint source pollution data in the YRD from 2010 to 2020. The specific calculation steps of the model are as follows.

**(1) Calculate the order ratio of the original data X (0):**

$$\lambda^{(0)}(t) = \frac{x^{(0)}(t-1)}{x^{(0)}(t)} \tag{5}$$

In Eq (5), t equals 1, 2, 3,. . .,n, if $\lambda^{(0)}(t) \in \left( e^{\frac{-2}{n+1}}, e^{\frac{2}{n+1}} \right)$, then the grey prediction model can be established from the original data.

**(2) Establish differential equation:**

$$x^{(0)}(k+1) = a[-0.5(x^{(1)}(k) + x^{(1)}(k+1))] + u \tag{6}$$

$$\text{Order } y = (x^{(0)}(2), x^{(0)}(3), x^{(0)}(4), \ldots x^{(0)}(N))^{(T)}, \hat{a} = \begin{pmatrix} a \\ u \end{pmatrix}$$

$$B = \begin{pmatrix} -0.5[x^{(1)}(2) + x^{(1)}(1)] & 1 \\ \vdots & \vdots \\ -0.5[x^{(1)}(N) + x^{(1)}(N-1)] & 1 \end{pmatrix}, \text{ then } \hat{a} = (B^T B)^{-1} B^T Y_n$$

In Eq (6), *a* and u are parameters.

**(3) Solving differential equations:**

$$\hat{x}^{(1)}(k+1) = \left[ x^{(1)}(1) - \frac{\hat{u}}{\hat{a}} \right] e^{-\hat{a}t} + \frac{\hat{u}}{\hat{a}} \tag{7}$$

In Eq (7), if k equals 1, 2, 3,. . .N-1, Then the obtained value is the fitting value, if k is bigger than or equal to N, then the obtained value is the predicted value.

## Results

### Time-series characteristics of agricultural pollution and carbon emissions

Table 4 shows the nonpoint source pollution emissions of various pollution sources in the YRD, and the trend of its emissions is similar to that found by Qiu et al. [59]. From 2010 to 2020, the COD from livestock and poultry farming in the YRD showed an upward and then downward trend, while COD emissions from aquaculture showed an upward trend. COD emissions from livestock farming are greater than those from aquaculture. Under the combined effect of the two activities, total agricultural COD emissions dropped from 1,130,120 tons in 2010 to 908,460 tons in 2020, with the lowest agricultural COD emissions of 798,660 tons in 2019, which is probably due to the significant drop in livestock farming in 2019 caused by swine fever, thus reducing agricultural COD emissions in that year.

From 2010 to 2020, the trends of agricultural TP emissions and TN emissions in the YRD were the same, and both emissions came from the same sources, namely, the plantation

**Table 4. Amount of agricultural nonpoint source pollution in the YRD, from 2010 to 2020.**

| Year | COD (10 kilo-tons) | | TN (10 kilo-tons) | | | TP (10 kilo-tons) | | |
|------|------------------|-------------|------------------------|------------------|-------------|------------------------|------------------|-------------|
| | Livestock Farming | Aquaculture | Agricultural fertilizers | Livestock farming | Aquaculture | Agricultural fertilizers | Livestock farming | Aquaculture |
| 2010 | 80.118 | 32.894 | 81.070 | 5.403 | 2.758 | 12.971 | 1.544 | 0.444 |
| 2011 | 83.630 | 34.274 | 84.388 | 5.647 | 2.882 | 13.502 | 1.597 | 0.462 |
| 2012 | 84.740 | 35.258 | 84.313 | 5.844 | 2.962 | 13.490 | 1.649 | 0.475 |
| 2013 | 84.819 | 36.158 | 84.349 | 5.994 | 3.032 | 13.496 | 1.644 | 0.486 |
| 2014 | 81.746 | 37.386 | 84.099 | 6.035 | 3.153 | 13.456 | 1.575 | 0.505 |
| 2015 | 79.503 | 38.189 | 83.198 | 6.090 | 3.246 | 13.312 | 1.547 | 0.521 |
| 2016 | 75.547 | 38.650 | 80.826 | 6.038 | 3.322 | 12.932 | 1.465 | 0.533 |
| 2017 | 72.296 | 37.358 | 78.782 | 5.942 | 3.171 | 12.605 | 1.395 | 0.509 |
| 2018 | 51.652 | 36.690 | 76.290 | 3.926 | 3.150 | 12.206 | 1.094 | 0.505 |
| 2019 | 43.330 | 36.536 | 73.539 | 3.806 | 3.161 | 11.766 | 1.019 | 0.507 |
| 2020 | 53.768 | 37.078 | 71.187 | 4.403 | 3.212 | 11.390 | 1.172 | 0.515 |

industry. Although TN and TP emissions from the farming industry show a trend of rising and then falling, the degree of fluctuation is not large, and the emissions are small. TN emissions from agricultural fertilizers account for approximately 90.445% of agricultural TN emissions. TN from agricultural fertilizers showed a decreasing trend after 2011, while TN emissions from livestock farming and aquaculture showed small fluctuations, with a turning point in 2015–2016. TN emissions from livestock farming and aquaculture peaked at 60,900 tons and 33,220 tons in 2015 and 2016, respectively. TP emissions from agricultural fertilizers accounted for approximately 86.958% of agricultural TP emissions, TP emissions from agricultural fertilizers showed a decreasing trend after 2011, and TP emissions caused by livestock farming and aquaculture showed a trend of first rising and then decreasing, peaking at 16,490 and 5,330 tons in 2012 and 2016, respectively. Notably, the nonpoint source pollution emissions calculated in this study are larger than the results of Wang et al. [60]. The main reason for the difference is the different nonpoint source pollution emissions from cultivation.

Table 5 shows the carbon emissions of agricultural emission sources in the YRD from 2010 to 2020. Agricultural carbon emissions calculated in this study are lower than those found by

**Table 5. Carbon emissions from agriculture in the YRD, from 2010 to 2020.**

| Year | Plantation (10 kilo-tons) | | | | | | Aquaculture (10 kilo-tons) | | |
|------|--------------------------|--------------|----------------------------|--------------------|------------------|----------------------------|---------|--------|--------|
| | Agricultural fertilizers | Pesticide use | Agricultural plastic film | Agricultural diesel | Farmland tilling | Agricultural land irrigation | Pig | Cow | Sheep |
| 2010 | 660.721 | 134.732 | 144.292 | 215.834 | 6.062 | 230.266 | 173.253 | 93.338 | 53.088 |
| 2011 | 687.759 | 135.438 | 153.159 | 224.050 | 6.260 | 239.336 | 185.013 | 96.222 | 53.783 |
| 2012 | 687.149 | 132.899 | 159.571 | 226.988 | 6.247 | 242.941 | 186.508 | 97.206 | 55.540 |
| 2013 | 687.445 | 131.548 | 164.066 | 231.880 | 6.248 | 260.780 | 184.741 | 98.186 | 57.038 |
| 2014 | 685.406 | 127.095 | 165.738 | 233.095 | 6.170 | 263.868 | 170.123 | 97.365 | 59.972 |
| 2015 | 678.064 | 123.843 | 161.816 | 236.282 | 6.190 | 264.666 | 160.585 | 95.932 | 63.404 |
| 2016 | 658.731 | 116.245 | 160.961 | 237.895 | 6.161 | 276.709 | 148.405 | 92.781 | 55.972 |
| 2017 | 642.076 | 109.927 | 161.096 | 237.435 | 6.040 | 284.160 | 140.863 | 89.630 | 54.641 |
| 2018 | 621.763 | 104.017 | 161.537 | 235.514 | 5.893 | 288.771 | 121.371 | 53.282 | 37.612 |
| 2019 | 599.343 | 100.792 | 161.652 | 230.272 | 5.879 | 281.966 | 77.977 | 52.032 | 37.529 |
| 2020 | 580.177 | 92.780 | 160.473 | 229.687 | 5.842 | 293.010 | 119.110 | 57.819 | 39.759 |

Tian et al. [61] and closer to those found by Hu et al. [62]. The main reason is that the minimum carbon emission sources selected in each study are different; the above studies take the provincial area as the research object, while this study takes the municipal areas of the YRD as the research object, so there are differences in the selection of indicators. From 2010 to 2020, total agricultural carbon emissions in the YRD show a trend of rising and then falling, Agricultural carbon emissions dropped from 17,115,900 tons in 2010 to 15,786,600 tons in 2020.

Carbon emissions from each emission source in descending order are agricultural fertilizer, agricultural irrigation, agricultural diesel, pigs, agricultural plastic film, pesticide use, cattle, sheep and farm tillage. Similar to the findings of Chen et al. [58], carbon emissions from the farming sector account for 83.320% of total carbon emissions from agriculture, much larger than those from the farming sector, of which fertilizer is the largest source. Carbon emissions from fertilizer have exceeded those from the farming sector, reaching a peak of 6,874,450 tons in 2013. However, since 2013, it has shown an annual trend of decreasing carbon emissions. Agricultural irrigation and agricultural diesel are the second and third largest sources of emissions from the farming sector, respectively, after fertilizers, emitting more than 2,000,000 tons. Carbon emissions from irrigation on farmland show a year-on-year increase, while those from diesel on farmland show an increase followed by a decrease, and those from tillage on farmland are the lowest, with a maximum fluctuation of no more than 5,000 tons. The use of pesticides has shown a year-on-year decrease since 2011. Carbon emissions from farming have come mainly from pig farming, which has shown a decreasing trend since 2011, except for 2019, when a significant drop in pig farming led to a decrease in carbon emissions. Carbon emissions from cattle and sheep farming show a downward trend after peaking at 981,860 tons and 634,040 tons in 2013 and 2015, respectively.

## Decoupling agricultural pollution and carbon reduction from economic growth

As shown in Fig 1, the general trend of decoupling between agricultural carbon emissions, nonpoint source pollution and agricultural economic development in the YRD from 2011 to 2020 is that the degree of decoupling is gradually increasing and the degree of coordination is increasing but with phased characteristics. The incompatibility between agricultural pollution and the economy is concentrated mainly in 2011, 2014 and 2020.

Specifically, in 2011, there were six cities with negative decoupling between agricultural carbon emissions and agricultural economic development, among which Lianyungang, Nantong, Suqian, Yancheng and Yangzhou showed strong negative decoupling, indicating that the development of the agricultural economy in central and northern Jiangsu was accompanied by increased agricultural carbon emissions in 2011, and there was much room for the development of low-carbon agriculture. In 2014, five cities showed strong negative decoupling, namely, Bozhou, Ma'anshan, Wuhu, Nantong and Zhenjiang.

In 2011, six cities showed strong negative decoupling between agricultural COD emissions and agricultural economic development, namely, Lianyungang, Nantong, Suqian, Tàizhou and Yangzhou. In 2014, the number of cities with strong negative decoupling decreased, but the number of cities with negative decoupling increased, and compared to 2011, the degree of negative decoupling between agricultural COD and economic development deepened. Notably, from 2014 to 2019, the degree of coordination between agricultural COD and economic development in the YRD continued to improve, but most cities in the YRD showed negative decoupling in 2020.

The degree of decoupling of agricultural TN and TP from the agricultural economy remained the same. In 2010, the negative decoupling cities were mainly Suqian and

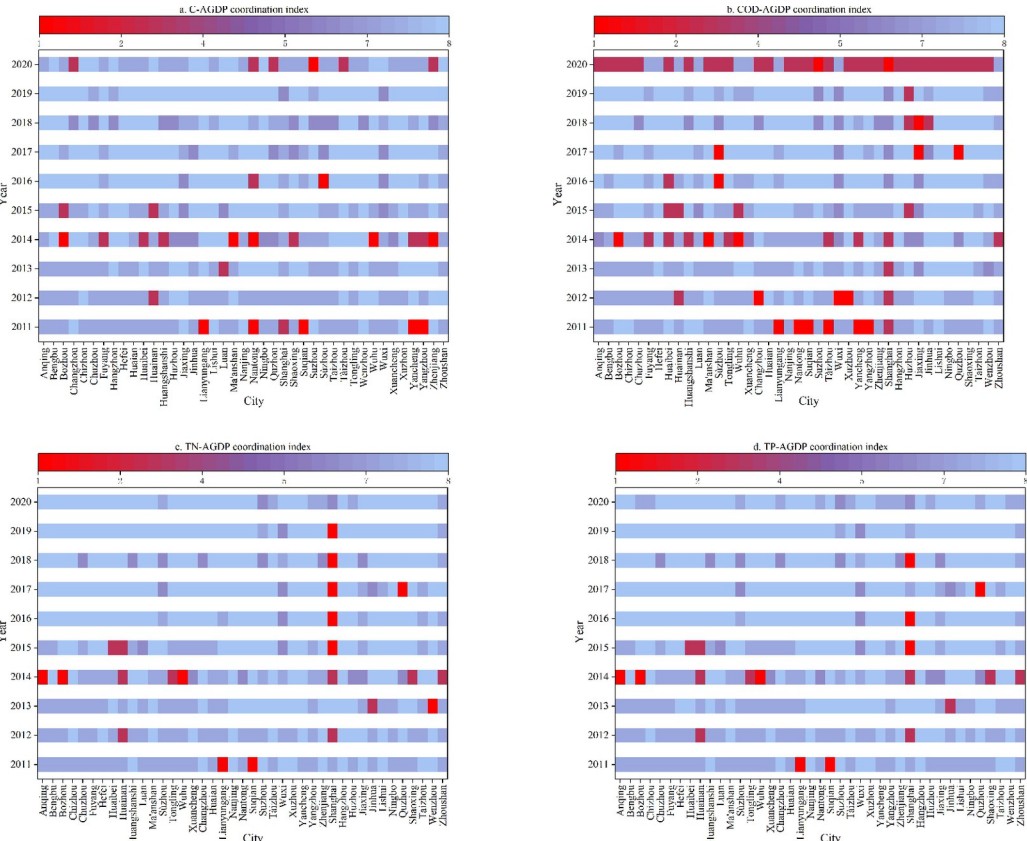

**Fig 1. Decoupling agricultural pollution and carbon reduction from economic growth in the YRD.**

Lianyungang. Eight cities showed negative decoupling in 2014, among which Anqing, Bozhou and Wuhu were in a strong negative decoupling state. Shanghai's agricultural TN and TP emissions and agricultural economy were in a state of negative decoupling for most of the period from 2014 to 2019, indicating that its agricultural economy developed weakly during this period, but pollution was stronger, and agricultural TN and TP emissions and economic development became unsynchronized.

## Agricultural pollution-carbon emission-economic growth change rate

The analysis in the previous section shows that pollution reduction, carbon reduction and agricultural economic development were relatively uncoordinated in 2011, 2014 and 2020. To further explore the decoupling of agricultural pollution and economic development in these years, the article measured the agricultural carbon, COD, TN, TP and agricultural economic change rates in the YRD in 2011, 2014 and 2020. The results are shown in **S2 Table** in the supporting information. The spatial pattern of the agricultural nonpoint source pollution reduction-carbon reduction-economic growth decoupling relationship changed significantly in 2011, 2014 and 2020.

Most of the YRD experienced rapid economic growth in 2011, and the rate of economic change was much greater than the rate of change in agricultural nonpoint source pollution and carbon emissions. The rates of change in agricultural COD emissions in Lianyungang, Yancheng, Nantong and Taizhou were greater than that in economic growth and other agricultural pollution emissions. The rate of change in agricultural carbon, TN and TP emissions

in Suqian were comparable and much greater than economic and agricultural COD growth, forming a dissonance with economic development. The negative decoupling of agricultural carbon emissions from the economy in Yangzhou is due mainly to the significant increase in carbon emissions. In 2011, the incongruence between agricultural nonpoint source pollution-carbon emissions-economic growth was concentrated mainly in the coastal and northern cities of Jiangsu, while the extreme incongruence between economic development and pollution emissions has not yet occurred in Zhejiang and Anhui.

In 2014, the negative decoupling of carbon emissions from the economy in Nantong and Zhenjiang was due mainly to the significant increase in carbon emissions, while the strong negative decoupling of carbon emissions and COD emissions from the economy in Bozhou, Ma'anshan and Wuhu was due mainly to the slow development of the agricultural economy. The strong negative decoupling between TN and TP and agricultural economic growth in Bozhou, Anqing and Wuhu was due mainly to the significant increase in TN and TP emissions.

In 2020, the agricultural COD in most cities entered a negative decoupling state with economic growth, becoming a major threat to agricultural development, which is due mainly to the significant increase in the amount of agricultural COD. Suzhou's agricultural carbon emissions are still in a strong negative decoupling state with economic development, which is due mainly to the slowdown of the agricultural economy; agricultural TN and TP are already in a decoupling state with economic growth, posing less of a threat to the environment and economy.

## Synergistic effects between agricultural pollution and carbon reduction

The synergy coefficients between agricultural pollution and carbon reduction in the YRD cities from 2011 to 2020 are represented in **Figs 2** and **3**. The synergistic effect between agricultural carbon and agricultural COD reduction is shown as an overall synergy, and the nonsynergistic cities are distributed in Jiangsu and Zhejiang Provinces.

Specifically, Lianyungang, Wenzhou, and Nantong showed serious nonsynergy between agricultural carbon and COD emission reduction in 2014 and 2016. Notably, these cities are important coastal port cities. In addition, the years in which agriculture exhibited a greater carbon emission reduction effect than COD emission reduction effect were concentrated in the first half of the study period, with representative cities being Ningbo and Yangzhou. An important reason for this observation may be the insufficient attention to rural pollution discharge in the previous years. After 2014, policies such as the Guidance on Improving the Rural Habitat Environment were introduced one after another, and rural sewage discharge was effectively managed.

Cities with uncoordinated agricultural carbon, TN and TP emission reduction effects are concentrated in Anhui Province and Jiangsu Province, and some of them are located in the Hang-Jia-Hu Plain area of Zhejiang Province. In regions with developed agriculture and plantation production as the main agricultural industry, the synergy of agricultural carbon and TN and TP emission reduction effects need to be improved. Especially in recent years, as awareness of carbon emission reduction increases, the treatment of TN and other nonpoint source pollution should be strengthened accordingly to avoid losing sight of the other pollution.

## Predicted decoupling of agricultural pollution and carbon reduction from economic growth

The article used the gray prediction model GM (1.1) to predict the decoupling relationship between agricultural pollution reduction and economic growth in the YRD from 2021 to 2030. The BP neural network model was combined to train the prediction results with unqualified

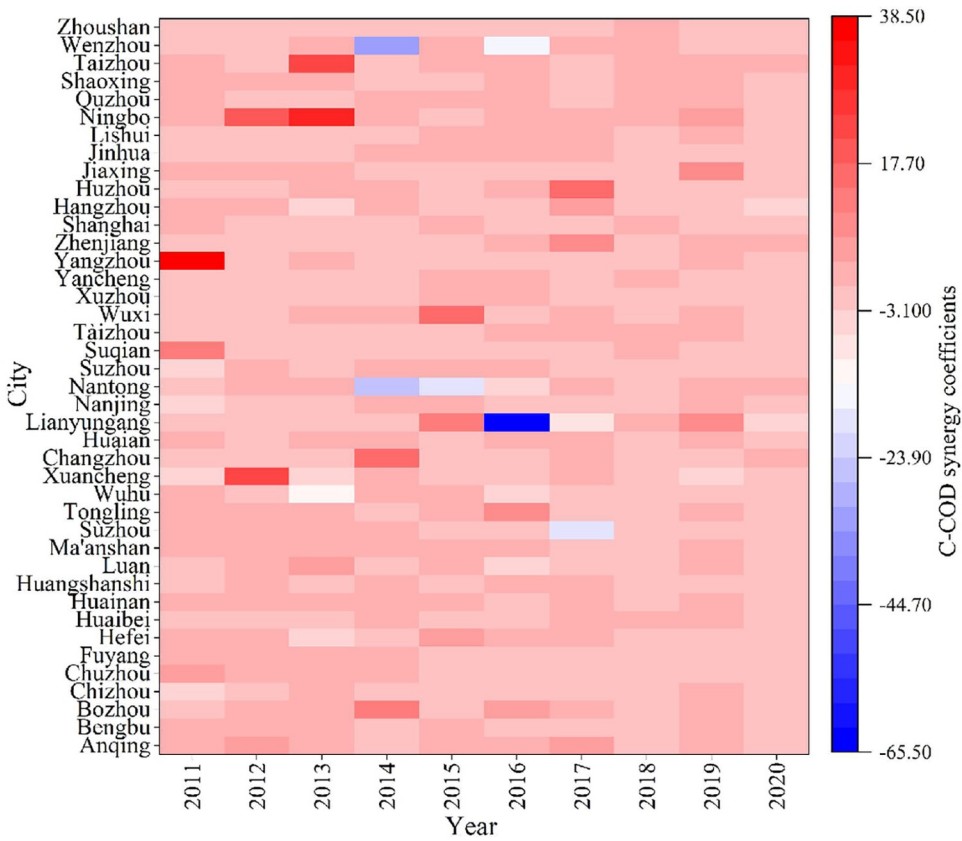

**Fig 2. Characteristics of synergistic effect of COD and carbon reduction in agriculture.**

accuracy until they were qualified [63, 64]. As shown in **Fig 4**, in general, the coherent relationship between agricultural pollution emissions and economic growth in the YRD continues to grow, with most cities showing decoupling during the forecast period and only some years showing negative decoupling, namely, in Anhui and Jiangsu provinces, which are two large agricultural provinces.

Specifically, in 2021, the agricultural carbon reduction effect was negatively decoupled from economic growth in Chuzhou, Fuyang, Huainan, Huangshan, Suzhou and Wuhu, the

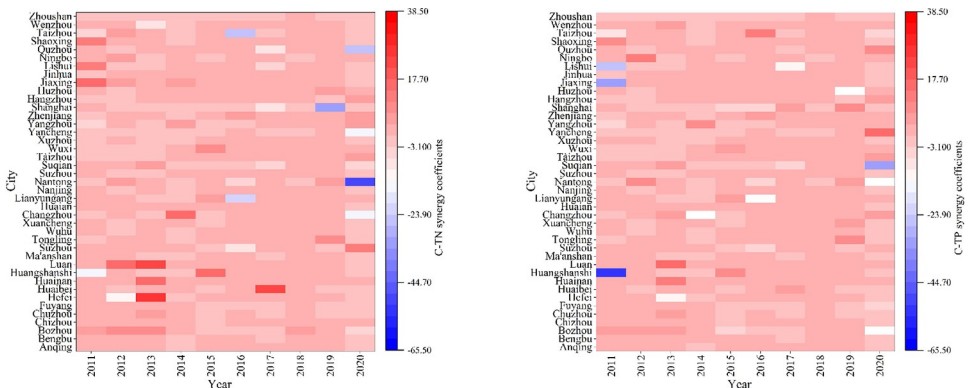

**Fig 3. Characteristics of synergistic effect of TN/TP and carbon reduction in agriculture.**

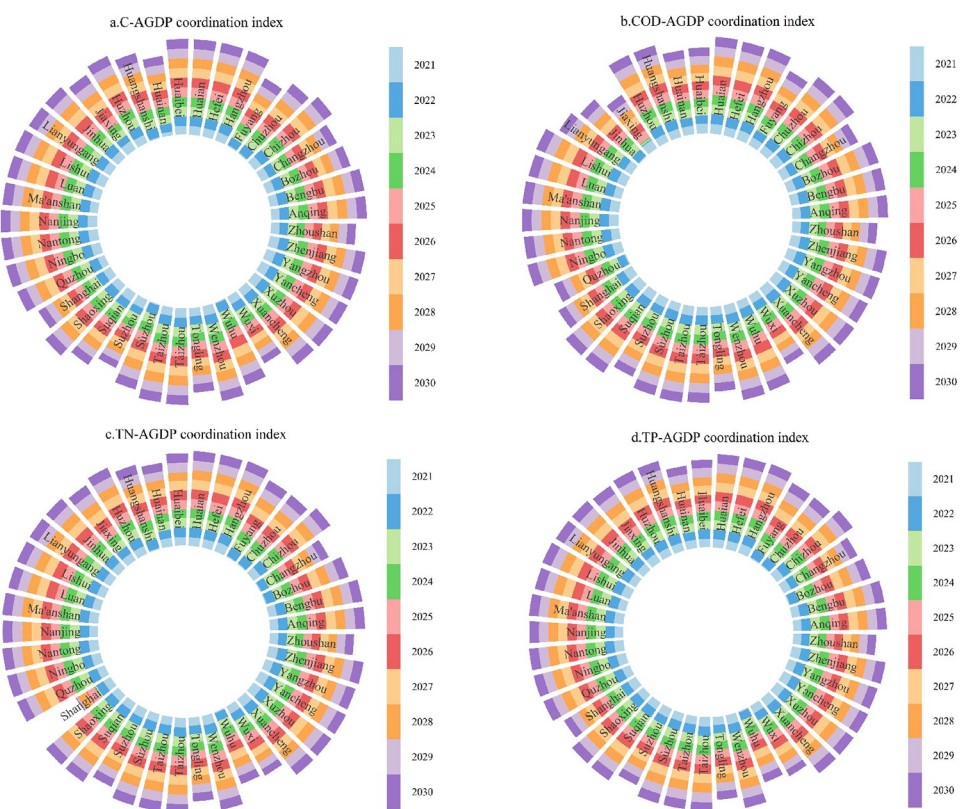

**Fig 4. Predicted decoupling of agricultural pollution and carbon reduction from agricultural economic growth.**

agricultural COD reduction effect was negatively decoupled from economic growth in Jinhua and Quzhou, the agricultural TN reduction effect was negatively decoupled from economic growth in Bozhou, Changzhou, Chuzhou, Fuyang and Wuhu, and the agricultural TP emission reduction effects were negatively decoupled from economic growth. In addition, the relationship between agricultural COD emission reduction effects and economic growth in Jiaxing from 2022 to 2024 is negatively decoupled, and the relationship between TN emission reduction effects and economic growth in Shanghai is negatively decoupled in 60% of the years, due mainly to the slow development of agriculture in these two regions and the forecasted negative growth in agricultural GDP.

## Discussion

The results show that in the past ten years, the distribution pattern of the main sources of agricultural nonpoint source pollution in the YRD has changed, and the total amount of pollutant discharge has decreased. The main contribution is the reduction in agricultural COD emissions. According to Zou and Wang et al. [60, 65], agricultural COD, TN and TP come mainly from livestock and poultry breeding. Therefore, to suppress the growth of agricultural nonpoint source pollution, we should focus on reducing livestock and poultry breeding pollution. During the research period, total agricultural carbon emissions in the YRD were high, showing a trend of first rising and then falling, similar to the national agricultural carbon emissions according to the findings of Liu et al. [38], but the reduction was not obvious. In addition, this study shows that the average growth rate of agricultural economy decreases from 11.14% in 2011 to 5.60% in 2020, and the average growth rates of carbon, COD emissions decrease from

4.74%, 5.18% in 2011 to 1.50%, 12.44% in 2020, the focus of agricultural environmental management is still to reduce carbon and COD emissions in Yangtze River Delta. There is a more concentrated negative decoupling between agricultural pollution and economic growth in the YRD in individual years. This was consistent with the findings for eastern and central China of Liu et al. [66]. They explained that it may be due to the depletion of agricultural resources and the decline in output levels. We agree with the above view and believe that with the rapid development of secondary and tertiary industries, the loss of rural population and backward agricultural production has caused this phenomenon. In addition, This study concluded that the agricultural COD emissions originate mainly from livestock and aquaculture, and the livestock and aquaculture emission coefficients of the YRD cities are high in China; due mainly to the impact of swine fever in 2019 and COVID-19, the rapid increase in the number of livestock and poultry breeding in the YRD in 2020 compared with the previous year caused a substantial increase in agricultural COD emissions, but the agricultural economy has not recovered to a corresponding degree, resulting in a lack of coordination between pollution emissions and the development of the agricultural economy. Therefore, this study makes policy recommendations in the following two aspects.

## Recommendations for the green economy development of agriculture

The integration of the three industries should be accelerated, thereby promoting the green development of agriculture. The latter cannot be achieved without a large amount of capital investment, and the government can increase support for capital policies; for example, for the northern Anhui and northern Jiangsu regions, which are located in large agricultural provinces but relatively economically backward areas, promoting green financial development can not only improve resource utilization efficiency and reduce environmental pollution, but also promote local economic development [67, 68]. In November 2021, the Implementation Plan for Green Financial Development in the YRD Ecological Green Integrated Development Demonstration Zone was officially issued, which provides great support for the green development of agriculture in the YRD and can further explore additional green financial measures to provide policy support for balancing economic growth and environmental protection. In addition, in terms of agricultural energy, the YRD region can strengthen the use of new energy sources, especially in coastal areas such as Lianyungang, Yancheng and Nantong, energy consumption is high and agricultural eco-efficiency is low, should strive to control pollution emissions at the source to achieve the dual carbon goal [69]. With the rapid development of digital industries in the YRD, boosting the digital economy can effectively promote the green and low-carbon development of the city, and increasing investment in R&D can reduce agricultural pollution and improve the production environment [51, 70]. With the deep integration of one, two, and three industries, advanced digital technologies and production techniques are now widely used in agricultural production as well as pollution prevention and control; for example, artificial intelligence and remote sensing have played an important role in pollutant monitoring and prevention [71, 72]. Therefore, in the future, advanced technologies can be further applied to agriculture, and investments in agricultural science and technology innovation can be increased to promote rapid agricultural economic growth from the production side and reduce environmental problems caused by agricultural development and achieve clean agricultural production from the pollution side [73, 74].

## Recommendations for agricultural environmental management

Promoting the combination of the market and the government, strengthening regional environmental synergy management. First, to promote coordinated economic and environmental

development, the Chinese government has taken a series of initiatives, including the promotion of third-party management of environmental pollution in China. As a market-based policy tool, it has broad application prospects in the YRD region, which has a high degree of marketization. The current degree of development of third-party treatment of environmental pollution in the YRD region is at a leading level nationwide [75], but in practice, problems such as inadequate market mechanisms and asymmetric market information persist. Especially for cities with good urban economic development but poor agricultural environmental management, such as Shanghai and Jiaxing, future measures such as strengthening the monitoring of rural land quality and water environment and further stimulating market dynamics through PPP and other forms can be taken. Second, agriculture should be included in the scope of the carbon market as soon as possible by taking advantage of the relatively low cost of agricultural emission reduction [76], which can not only accelerate agricultural carbon emission reduction but also help to promote sustainable emission reduction in agricultural nonpoint source pollution. Finally, environmental management is not the responsibility of a particular region. It can be seen that Jiangsu and Anhui, both of which are large agricultural provinces, are in a disadvantageous position in terms of agro-ecological environment, especially in the regions bordering the two provinces, where the agricultural environment and the economy show a negative decoupling of the majority of cities. At the same time, the relationship between the mountains and water and the economy provides geographical conditions for the comprehensive management of the rural environment in many places. Local governments at all levels can implement advanced management techniques, promote intersectoral as well as intergroup cooperation, and give full play to both the spillover effect of technology and knowledge synergy [77] to achieve collaborative agro-environmental governance [78].

## Conclusions

This article analyzes the spatial and temporal partitioning of agricultural pollution in the YRD by measuring agricultural nonpoint source pollution and carbon emissions. It uses the Tapio decoupling model and the gray prediction GM (1,1) model to analyze and predict the relationship between pollution reduction and carbon reduction and agricultural economic development in the YRD. The conclusions are as follows:

(1) Compared to 2010, total agricultural nonpoint source pollution and carbon emissions in the YRD in 2020 were significantly lower, with small fluctuations during this period. The focus of agricultural COD emission reduction should shift from livestock and poultry farming to a joint reduction in livestock and aquaculture, strengthening agricultural emission reduction efforts in coastal cities with a focus on agricultural carbon emission reduction in the plantation sector, while being aware of the significant increase in carbon emissions resulting from the recovery of the farming sector.

(2) In the YRD, the agricultural pollution reduction and carbon reduction effects and economic growth are in a decoupled state, and a few regions show negative decoupling, mainly in 2011, 2014 and 2020, where the agricultural C, TN and TP reduction effects were better coordinated with economic growth, and the agricultural COD reduction effects were in greater conflict with economic growth. According to the prediction results, except for 2021, when the degree of coordination between the agricultural pollution reduction and carbon reduction effect and economic growth is poor, both show a better decoupling state in all other years and a better decoupling effect. Some regions face severe negative decoupling between agricultural COD and TN emissions and economic growth, especially Jiaxing and Shanghai, and the main reason may be the low potential for agricultural economic development in the two regions.

(3) This study takes 41 cities in the YRD as research objects, considering the agricultural development status of each city. It considers the relationship between the agricultural economy and the environment in terms of both carbon emissions and nonpoint source pollution, which has been rare in previous studies. However, due to the limited data availability, the carbon emissions calculated in this study ignore the implied carbon emissions of agricultural production, and the index system may not fully reflect the pollution status of agricultural development. In addition, the large randomness of some original series in the gray prediction model impacts the prediction results; even if the application of the neural network model is added, it cannot accurately reflect the development status of the agricultural economy and environment in the next ten years. In future research, we will address the shortcomings of this study to better understand the role of agriculture in achieving the dual carbon target.

## Supporting information

**S1 Table. Decoupling agricultural pollution and carbon reduction from economic growth.**
(XLSX)

**S2 Table. Agricultural pollution-carbon emission-economic growth change rate.**
(XLSX)

**S3 Table. The synergy coefficients between agricultural pollution and carbon reduction in the Yangtze River Delta cities from 2011 to 2020.**
(XLSX)

**S4 Table. Predicted decoupling of agricultural pollution and carbon reduction from economic growth.**
(XLSX)

## Author Contributions

**Data curation:** Ruxue Yuan.

**Investigation:** Ruxue Yuan.

**Methodology:** Ruxue Yuan, Fanbin Kong.

**Software:** Ruxue Yuan.

**Writing – original draft:** Ruxue Yuan.

**Writing – review & editing:** Ruxue Yuan, Caiyao Xu, Fanbin Kong.

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
