## [Decision Letter · Decision Letter 0]

8 Aug 2022

PONE-D-22-14888Decoupling agriculture pollution & carbon reduction from economic growth in the Yangtze River Delta, ChinaPLOS ONE

Dear Dr. Xu,

Thank you for submitting your manuscript to PLOS ONE. After careful consideration, we feel that it has merit but does not fully meet PLOS ONE’s publication criteria as it currently stands. Therefore, we invite you to submit a revised version of the manuscript that addresses the points raised during the review process.

Based on the reviewer's recommendation, the manuscript needs a MINOR REVISION before it can be reconsidered for publication.

We look forward to receiving your revised manuscript.

Kind regards,

Nor Adilla Rashidi, Ph.D.

Academic Editor

PLOS ONE

Journal Requirements:

2. We note that Figures 1 and 3 in your submission contain [map/satellite] images which may be copyrighted. All PLOS content is published under the Creative Commons Attribution License (CC BY 4.0), which means that the manuscript, images, and Supporting Information files will be freely available online, and any third party is permitted to access, download, copy, distribute, and use these materials in any way, even commercially, with proper attribution. For these reasons, we cannot publish previously copyrighted maps or satellite images created using proprietary data, such as Google software (Google Maps, Street View, and Earth). For more information, see our copyright guidelines: http://journals.plos.org/plosone/s/licenses-and-copyright.

   1. You may seek permission from the original copyright holder of Figures 1 and 3 to publish the content specifically under the CC BY 4.0 license.  

“This research was supported by the Zhejiang Provincial Natural Science Foundation of China (Grant No. Z22D010686), the National Natural Science Foundation of China (Grant No. 41871083), Zhejiang Soft Science Research Program of China (Grant No. 2022C35104), Research Development Fund of Zhejiang A&F University (Grant No. 2020FR066), and the Special Project of Cultivating Leading Talents in Philosophy and Social Science of Zhejiang Province (Grant No. 21YJRC12-2YB and 21YJRC2ZD)”

Please respond by return e-mail so that we can amend your financial disclosure and competing interests on your behalf.

Reviewers' comments:

Reviewer's Responses to Questions

**Comments to the Author**

1. Is the manuscript technically sound, and do the data support the conclusions?

Reviewer #1: Yes

Reviewer #2: Yes

2. Has the statistical analysis been performed appropriately and rigorously? 

Reviewer #1: Yes

Reviewer #2: Yes

3. Have the authors made all data underlying the findings in their manuscript fully available?

Reviewer #1: Yes

Reviewer #2: Yes

4. Is the manuscript presented in an intelligible fashion and written in standard English?

Reviewer #1: Yes

Reviewer #2: No

5. Review Comments to the Author

Reviewer #1: The article analyzes the spatial and temporal partitioning of agricultural pollution in the Yangtze River Delta by measuring agricultural non-point source pollution and carbon emissions and uses the Tapio decoupling model and the grey prediction GM (1,1) model to analyze and predict the relationship between pollution reduction and carbon reduction and agricultural economic development in the Yangtze River Delta. I firmly believe that the authors have done a great deal of hard work. However, there are still some areas that can be improved:

(1) The Introduction part should start from the phenomena and problems in practice and lead to the research problem. Some real-world cases should be mentioned and discussed to enhance the research motivation.

(2) The literature review is insufficient and does not explain the shortcomings of previous research and the innovation of this paper. The innovation of this paper and the contribution made by previous studies have not been clearly expressed.

(3) What are the characteristics of the Yangtze River Delta, why do the authors use YRD as a case study, and what’s the problem of the relationship between pollution reduction and carbon reduction, and agricultural economic development?

(4) The authors should analyze the results of this paper, as well as the differences from other research results, and introduce some inspiring findings and viewpoints of this paper.

(5) What are the managerial insights/policy implications of this study? Compared with available literature, what are the theoretical contributions and application values of this study? It is suggested to enhance the corresponding discussions in the conclusion part.

(6) It is better if the authors could introduce the limitations of the study and future studies in the conclusion section. The following literature should be helpful for your research: Decoupling economic growth from water consumption in the Yangtze River Economic Belt, China. And Coordination of the Industrial-Ecological Economy in the Yangtze River Economic Belt, China.

Reviewer #2: Following changes should be done before the consideration of this paper:

1- Need to replace the word "&" with "and".

2- The abstract is presented poorly, i suggest authors to make it more constructive.

3- The introduction part need to expand little more, and clearly demonstrate the major aim of this study that whats novel in it and how it is going to contribute in the existing literature.

4- In the materials and methods, need to add the url link of data sources. Further, need to improve the quality of Figures 1 and 4.

5- The findings of the Tables 4 & 5 need more explanation by comparing it with previous published studies.

6- Need to expand discussion little more.

7- The Conclusions should be Conclusion and Policy recommendations by recommending some policies in the light of this study findings. Further, need to add limitations and future research directions.

8- The language of the paper should be check with the help of a native English speaker to improve its quality.

6. PLOS authors have the option to publish the peer review history of their article (what does this mean?). If published, this will include your full peer review and any attached files.

Reviewer #1: No

Reviewer #2: No

---

## [Author Response · Author response to Decision Letter 0]

16 Nov 2022

Modification instructions

(Manuscript number: PONE-D-22-14888; Modify date: August 26, 2022)

Dear Reviewers/Editorial Board Members:

Thank you very much for your comments on our paper "Decoupling agriculture pollution & carbon reduction from economic growth in the Yangtze River Delta, China ", your comments are very valuable to us. We have revised the paper according to the review comments, and we would like to explain each of the changes.

Reviewer #1 Review Comments

External review comments1: (1) The Introduction part should start from the phenomena and problems in practice and lead to the research problem. Some real-world cases should be mentioned and discussed to enhance the research motivation.

Response1: Thanks for the kind valuable advice and the expert's advice is very reasonable. The author revised the introduction section according to the experts' suggestions, adding the pollution problems existing in the reality of the Yangtze River Delta and enhancing the motivation of the study. The revision is as follows.

Since the reform and opening up, China's agricultural production conditions have continued to improve, providing strong support for the development of agricultural modernization, with an average annual growth rate of 4.6% in the gross agricultural product (AGDP) from 1978 to 2020 [1]. However, the development of the agricultural economy has generated agricultural pollution, and with the disappearance of the demographic dividend, environmental pollution and resource depletion have become increasingly prominent and counterproductive constraints on the development of the agricultural economy [2, 3]. Agricultural production is highly dependent on the environment and climate, and accurately characterizing the relationship between environmental pollution in agriculture and economic growth is an important topic that needs attention. Therefore, it is important to study the decoupling of agricultural pollution and carbon reduction from economic growth to achieve sustainable agricultural development. 

The current situation of agricultural surface pollution in China is very serious [4], and the total amount and intensity of carbon emissions from agricultural land use are increasing [5]. According to data from the Second National Pollution Census of China and the Food and Agriculture Organization of the United Nations, China's agricultural chemical oxygen demand (COD), total nitrogen (TN), and total phosphorus (TP) emissions reached 10,671,300 tons, 1,414,900 tons, and 212,000 tons, respectively, in 2017, accounting for half of all pollution emissions. Global greenhouse gas emissions from agricultural land exceeded 30% of total global anthropogenic greenhouse gas emissions, and China's agricultural source greenhouse gas emissions accounted for approximately 17% of total national greenhouse gas emissions [6]. In addition, agriculture is not only a "contributor" to carbon emissions, but also one of the sectors most vulnerable to climate change. As the global warming trend intensifies, agricultural production will become limited; thus, it is crucial to balance the relationship between economic development and agricultural carbon emissions and agricultural surface pollution. China proposed a "double carbon" goal at the 75th United Nations General Assembly and incorporated it into its ecological civilization system in 2020. In December 2020, the Central Economic Work Conference proposed that "we should continue to fight the battle against pollution and achieve the synergistic effect of pollution reduction and carbon reduction" and included this goal in the 14th Five-Year Plan for national economic and social development. The No. 1 Document of the Central Government in 2022 proposed strengthening the comprehensive management of agricultural nonpoint source pollution and promoting green development in agriculture and rural areas. Fertilizers, pesticides and agricultural films are the main sources of not only agricultural nonpoint source pollution but also greenhouse gas nitrous oxide emissions, so reducing the consumption of fertilizers, pesticides and agricultural films is a synergistic strategy to reduce pollution and carbon emissions [7]. Therefore, we can start with agricultural production behavior to pro-mote the decoupling of pollution and carbon reduction from economic growth and help achieve sustainable agricultural development.

According to calculations based on data published in the 2020 China Environmental Statistics Yearbook, in 2019, the amount of agricultural fertilizer applied in the Yangtze River Delta region reached 6,642,000 tons, the number of pesticides used reached 197,010 tons, the amount of plastic film used was 297,840 tons, and the amount of agricultural COD and agricultural ammonia nitrogen in wastewater discharge accounted for 8.856% and 22.672% of the country's total, respectively, posing a serious threat to the ecology of China's watersheds. Therefore, this study takes 41 municipalities in the Yangtze River Delta as the research object, uses the Tapio model to analyze the decoupling relationship between pollution reduction and carbon reduction and economic growth in the Yangtze River Delta from 2011 to 2020. It predicts the future decoupling state between the two variables from 2020 to 2030 by using a gray prediction model. The remainder of the article is organized as follows. Section II compares the literature on the subject terms of agricultural nonpoint source pollution and agricultural carbon emissions. Section III describes the study area overview, research methodology and data sources. Section IV presents the study results. Section V reports and analyzes the study results and gives policy recommendations, Section Ⅵ presented the conclusions and limitations of the study.

References

[1] Huang JK, Xie W, Sheng Y, Wang XB, Wang JX, Liu CF, et al. Trends of Global Agriculture and Prospects of China's Agriculture Toward 2050. Strategic Study of CAE. 2022;24(01):29-37.

[2] Li S, Gong Q, Yang S. Analysis of the Agricultural Economy and Agricultural Pollution Using the Decoupling Index in Chengdu, China. International Journal of Environmental Research and Public Health. 2019;16(21). doi: 10.3390/ijerph16214233. PubMed PMID: WOS:000498842000179.

[3] Wang XB, Chen CX. The Pressure from Economic Growth Target and the Green Development Quality of Manufacturing Industry——A Measurement and Econometric Analysis Based on GTFP. Journal of Macro-quality Research. 2021;9(03):50-69. doi: 10.13948/j.cnki.hgzlyj.2021.03.005.

[4] Li YY, L X, Meng C, Wu JS. Analysis of agricultural non-point source pollution issue in waters and technical strategy of comprehensive prevention and control in rural area of China. Research of Agricultural Modernization. 2021;42(02):185-97. doi: 10.13872/j.1000-0275.2021.0025.

[5] Li B, Wang CY, Zhang JB. Dynamic evolution and spatial spillover of China's agricultural net carbon sink. China Population Resources and Environment. 2019;29(12):68-76.

[6] Han JY, Qu JS, Xu L, Li HJ, Liu LN. The Spatial Effect of Agricultural Finance on Agricultural Greenhouse Gas Emission Intensity: An Empirical Analysis Based on the Spatial Durbin Model. Journal of Ecology and Rural Environment. 2021. 2021;37(11):1404-12. doi: 10.19741/j.issn.1673-4831.2020.0783.

[7] Su LY. Research on the Strategy of Integrating Carbon Peak and Carbon Neutrality into the Overall Layout of Ecological Civilization Construction. Environmental Protection. 2021;49(16):6-9. doi: 10.14026/j.cnki.0253-9705.2021.16.002

External review comments2: (2) The literature review is insufficient and does not explain the shortcomings of previous research and the innovation of this paper. The innovation of this paper and the contribution made by previous studies have not been clearly expressed.

Response2: Thanks for the kind valuable advice and the expert's advice is very reasonable. The author revised the literature review section based on expert suggestions, summarized the shortcomings of previous studies, and added research contributions to this paper. The modifications are as follows.

Existing literature: (1) Although a more unified methodological system has been developed for the accounting of agricultural surface source pollution and agricultural carbon emissions, consistent results have not yet emerged due to different data sources and specific categories of accounting, and some results differ greatly. (2) The Central Economic Work Conference in December 2020 proposed that "we should continue to fight the battle against pollution and achieve synergistic effects of pollution reduction and carbon reduction". Studies on carbon reduction are still at the level of legal system, theoretical level and realization path, but there is a lack of corresponding empirical research and quantitative analysis. (3) The application of decoupling theory in the agricultural sector is relatively rare, especially in the study of the relationship between agricultural nonpoint source pollution and the agricultural economy. In addition, most studies take the provincial area as the basic unit or a specific city as the research object, lacking the concept of region.

Research contributions: (1) Drawing on the methods of previous studies, we measured agricultural surface source pollution and carbon emissions in the Yangtze River Delta with prefecture-level cities as the research objects, bridging the gap in previous studies. (2) The relationship between agricultural environmental pollution and economic growth from both agricultural nonpoint source pollution and carbon emissions is considered. (3) The relationship between agricultural pollution reduction and economic growth is analyzed from both temporal and spatial perspectives, providing ideas and a scientific basis for reducing three-dimensional pollution at the source and achieving synergistic management of the agricultural environment and climate. (4) The decoupling trend of agricultural pollution reduction and carbon reduction from economic growth for the next ten years is predicted, and policy suggestions are put forward, focusing on the use of agricultural technology and green finance to improve the efficiency of agricultural green production.

External review comments3: (3) What are the characteristics of the Yangtze River Delta, why do the authors use Yangtze River Delta as a case study, and what’s the problem of the relationship between pollution reduction and carbon reduction, and agricultural economic development?

Response3: Thanks for the kind questions. The author's responses to the review comments3 are as follows.

Reasons for using the Yangtze River Delta as the study area:

 (1) According to calculations based on data published in the 2020 China Environmental Statistics Yearbook, in 2019, the amount of agricultural fertilizer applied in the Yangtze River Delta region reached 6,642,000 tons, the number of pesticides used reached 197,010 tons, the amount of plastic film used was 297,840 tons, and the amount of agricultural COD and agricultural ammonia nitrogen in wastewater discharge accounted for 8.856% and 22.672% of the country's total, respectively, posing a serious threat to the ecology of China's watersheds.

(2) The Yangtze River Delta is an important production area for commercial foodstuffs and a pilot demonstration area for sustainable agricultural development, which is strategically important in promoting national agricultural development and maintaining food security. 

(3) The spread of agricultural pollutants will lead to serious soil and water pollution. The Yangtze River Delta is bordered by the Yellow Sea and the East China Sea, with a dense network of rivers and streams, and offers a geographical advantage for studying environmental pollution in agriculture.

The problem of the relationship between pollution reduction and carbon reduction, and agricultural economic development:

 (1) The development of the agricultural economy has generated agricultural pollution, and with the disappearance of the demographic dividend, environmental pollution and resource depletion have become increasingly prominent and counterproductive constraints on the development of the agricultural economy [2, 3]. Agricultural production is highly dependent on the environment and climate. 

(2) This study shows the agricultural pollution reduction and carbon reduction effects and economic growth are in a decoupled state, and a few regions show negative decoupling, mainly in 2011, 2014 and 2020, where the agricultural C, TN and TP reduction effects were better coordinated with economic growth, and the agricultural COD reduction effects were in greater conflict with economic growth.

References

[2] Li S, Gong Q, Yang S. Analysis of the Agricultural Economy and Agricultural Pollution Using the Decoupling Index in Chengdu, China. International Journal of Environmental Research and Public Health. 2019;16(21). doi: 10.3390/ijerph16214233. PubMed PMID: WOS:000498842000179.

[3] Wang XB, Chen CX. The Pressure from Economic Growth Target and the Green Development Quality of Manufacturing Industry——A Measurement and Econometric Analysis Based on GTFP. Journal of Macro-quality Research. 2021;9(03):50-69. doi: 10.13948/j.cnki.hgzlyj.2021.03.005.

External review comments4: (4) The authors should analyze the results of this paper, as well as the differences from other research results, and introduce some inspiring findings and viewpoints of this paper.

Response4: Thanks for the kind valuable advice and the expert's advice is very reasonable. The author added a comparison with the results of previous calculations in the results analysis section based on expert recommendations, which were modified as follows.

 Table 4 shows the non-point source pollution emissions of various pollution sources in the Yangtze River Delta, and the trend of its emissions is similar to that found by Qiu et al. [57]. From 2010 to 2020, the COD from livestock and poultry farming in the Yangtze River Delta showed an upward and then downward trend, while COD emissions from aquaculture showed an upward trend.

TP emissions from agricultural fertilizers accounted for approximately 86.958% of agricultural TP emissions, TP emissions from agricultural fertilizers showed a decreasing trend after 2011, and TP emissions caused by livestock farming and aquaculture showed a trend of first rising and then decreasing, peaking at 16,490 and 5,330 tons in 2012 and 2016, respectively. Notably, the surface source pollution emissions calculated in this study are larger than the results of Wang et al. [58]. The main reason for the difference is the different surface source pollution emissions from cultivation.

Table 5 shows the carbon emissions of agricultural emission sources in the Yangtze River Delta from 2010 to 2020. Agricultural carbon emissions calculated in this study are lower than those found by Tian et al. [59] and closer to those found by Hu et al. [60]. The main reason is that the minimum carbon emission sources selected in each study are different; the above studies take the provincial area as the research object, while this study takes the municipal areas of the Yangtze River Delta as the research object, so there are differences in the selection of indicators. From 2010 to 2020, total agricultural carbon emissions in the Yangtze River Delta show a trend of rising and then falling. Carbon emissions from each emission source in descending order are agricultural fertilizer, agricultural irrigation, agricultural diesel, pigs, agricultural plastic film, pesticide use, cattle, sheep and farm tillage. Similar to the findings of Chen et al. [56], carbon emissions from the farming sector account for 83.320% of total carbon emissions from agriculture, much larger than those from the farming sector, of which fertilizer is the largest source.

References

[56] Chen ST, Zhang KH, Zhang YW. Measurement and decoupling effects of agricultural carbon emission performance. STATISTICS & DECISION. 2021;37(22):85-8. doi: 10.13546/j.cnki.tjyjc.2021.22.019.

[57] Qiu WW, Zhong ZB, Li ZL. Agricultural Non-point Source Pollution in China: Evaluation, Convergence Characteristics and Spatial Effects. Chin Geogr Sci. 2021;31(3):571-84. doi: 10.1007/s11769-021-1200-1. PubMed PMID: WOS:000648287300013.

[58] Wang SR, Yang DW, Sun JH, Tang LH, Wang PJ, et al. Analysis on status and characteristics of agricultural non-point source pollution in China. Water Resources Protection. 2021;37(04):140-7+72.

[59] Tian Y, Wu HT. Research on Fairness of Agricultural Carbon Emissions in China’s Major Grain Producing Areas from the Perspective of Industrial Structure. Journal of Agrotechnical Economics. 2020;(01):45-55. doi: 10.13246/j.cnki.jae.2020.01.003.

[60] Hu WL, Zhang JX, Wang HL. Characteristics and Influencing Factors of Agricultural Carbon Emission in China. STATISTICS & DECISION. 2020;36(05):56-62.

External review comments5: (5) What are the managerial insights/policy implications of this study? Compared with available literature, what are the theoretical contributions and application values of this study? It is suggested to enhance the corresponding discussions in the conclusion part.

Response5: Thanks for the kind questions. The author put forward relevant policy recommendations in the discussion section of the article based on expert recommendations.

 Compared with the existing literature, this article has studied 41 prefecture-level cities in the Yangtze River Delta, a region of special demonstration significance, which is rare among existing studies. In addition, this paper explores the relationship between carbon reduction, pollution reduction and economic growth in agriculture, which fills the research gap.

Managerial insights/policy implications: 

(1) Recommendations for the green economy development of agriculture 

The integration of the three industries should be accelerated, thereby promoting the green development of agriculture. The latter cannot be achieved without a large amount of capital investment, and the government can increase support for capital policies; for example, promoting green financial development can improve resource utilization efficiency and reduce environmental pollution [63, 64]. In November 2021, the Implementation Plan for Green Financial Development in the Yangtze River Delta Ecological Green Integrated Development Demonstration Zone was officially issued, which provides great support for the green development of agriculture in the Yangtze River Delta and can further explore additional green financial measures to provide policy support for balancing economic growth and environmental protection. In addition, in terms of agricultural energy, the Yangtze River Delta region can strengthen the use of new energy sources, actively promote new energy sources such as photovoltaic and solar energy, and strive to control carbon emissions at the source to achieve the dual carbon goal [65]. With the rapid development of digital industries in the Yangtze River Delta, boosting the digital economy can effectively promote the green and low-carbon development of the city, and increasing investment in R&D can reduce agricultural pollution and improve the production environment [48, 66]. With the deep integration of one, two, and three industries, advanced digital technologies and production techniques are now widely used in agricultural production as well as pollution prevention and control; for example, artificial intelligence and remote sensing have played an important role in pollutant monitoring and prevention [67, 68]. Therefore, in the future, advanced technologies can be further applied to agriculture, and investments in agricultural science and technology innovation can be increased to promote rapid agricultural economic growth from the production side and to reduce environmental problems caused by agricultural development and achieve clean agricultural production from the pollution side [69, 70].

(2) Recommendations for agricultural environmental management

Promoting the combination of the market and the government, strengthening regional environmental synergy management. First, to promote coordinated economic and environ-mental development, the Chinese government has taken a series of initiatives, including the promotion of third-party management of environmental pollution in China. As a market-based policy tool, it has broad application prospects in the Yangtze River Delta region, which has a high degree of marketization. The current degree of development of third-party treatment of environmental pollution in the Yangtze River Delta region is at a leading level nationwide [71], but in practice, problems such as inadequate market mechanisms and asymmetric market information persist, and future measures such as strengthening the monitoring of rural land quality and water environment and further stimulating market dynamics through PPP and other forms can be taken. Second, agriculture should be included in the scope of the carbon market as soon as possible by taking advantage of the relatively low cost of agricultural emission reduction [72], which can not only accelerate agricultural carbon emission reduction but also help to promote sustainable emission reduction in agricultural surface source pollution. Finally, environmental management is not the responsibility of a particular region; the relationship between mountains and water and economic interconnection requires the Yangtze River Delta to achieve comprehensive rural environmental management. Each region should not only adopt differentiated governance measures according to local conditions but also cooperate with each other to achieve synergistic governance. As the integration of the Yangtze River Delta becomes a national strategy, interregional cooperation and resource sharing are increasing. Local governments at all levels can implement advanced management techniques, promote intersectoral as well as intergroup cooperation, and give full play to both the spillover effect of technology and knowledge synergy [73] to achieve collaborative agro-environmental governance [74].

References

[48] Wang Q, Zhang FY. Does increasing investment in research and development promote economic growth decoupling from carbon emission growth? An empirical analysis of BRICS countries. Journal of Cleaner Production. 2020;252. doi: 10.1016/j.jclepro.2019.119853. PubMed PMID: WOS:000516777200100

[63] Zhang S, Wu Z, Wang Y, Hao Y. Fostering green development with green finance: An empirical study on the environmental effect of green credit policy in China. Journal of Environmental Management. 2021;296. doi: 10.1016/j.jenvman.2021.113159. PubMed PMID: WOS:000685497300005.

[64] Fang Y, Shao ZQ. Whether Green Finance Can Effectively Moderate the Green Technology Innovation Effect of Heterogeneous Environmental Regulation. International Journal of Environmental Research and Public Health. 2022;19(6). doi: 10.3390/ijerph19063646. PubMed PMID: WOS:000776864100001.

[65] Zhang Y, Yu Z, Zhang J. Spatiotemporal evolution characteristics and dynamic efficiency decomposition of carbon emission efficiency in the Yellow River Basin. Plos One. 2022;17(3). doi: 10.1371/journal.pone.0264274. PubMed PMID: WOS:000780951300016.

[66] Wang L, Chen L, Li Y. Digital economy and urban low-carbon sustainable development: the role of innovation factor mobility in China. Environmental Science and Pollution Research. 2022. doi: 10.1007/s11356-022-19182-2. PubMed PMID: WOS:000759365600010.

[67] Liu W, Xu Y, Fan D, Li Y, Shao X-F, Zheng J. Alleviating corporate environmental pollution threats toward public health and safety: The role of smart city and artificial intelligence. Safety Science. 2021;143. doi: 10.1016/j.ssci.2021.105433. PubMed PMID: WOS:000690367600023.

[68] Wang P, Yao J, Wang G, Hao F, Shrestha S, Xue B, et al. Exploring the application of artificial intelligence technology for identification of water pollution characteristics and tracing the source of water quality pollutants. Science of the Total Environment. 2019;693. doi: 10.1016/j.scitotenv.2019.07.246. PubMed PMID: WOS:000489694700047.

[69] Yi X, Lin D, Li J, Zeng J, Wang D, Yang F. Ecological treatment technology for agricultural non-point source pollution in remote rural areas of China. Environmental Science and Pollution Research. 2021;28(30):40075-87. doi: 10.1007/s11356-020-08587-6. PubMed PMID: WOS:000528989000002.

[70] Liu SJ, Xu XL. The pollution halo effect of technology spillover and pollution haven effect of economic growth in agricultural foreign trade: two sides of the same coin? Environmental Science and Pollution Research. 2021;28(16):20157-73. doi: 10.1007/s11356-020-11786-w. PubMed PMID: WOS:000605568500037.

[71] Chen HJ, Si W. Third Party Govern of Environmental Pollution in the Yangtze River Delta Region: Current Situation, Problems and Suggestion. Environmental Protection. 2020;48(20):20-3. doi: 10.14026/j.cnki.0253-9705.2020.20.004.

[72] Jin SQ, Lin Y, Niu KY. Driving Green Transformation of Agriculture with Low Carbon: Characteristics of Agricultural Carbon Emissions and Its Emission Reduction Path in China. Reform. 2021;(05):29-37.

[73] Xu L, Zhou Z, Du J. An Evolutionary Game Model for the Multi-Agent Co-Governance of Agricultural Non-Point Source Pollution Control under Intensive Management Pattern in China. International Journal of Environmental Research and Public Health. 2020;17(7). doi: 10.3390/ijerph17072472. PubMed PMID: WOS:000530763300304.

[74] Hu WJ. Improve the Yangtze River Delta regional pollution prevention and control collaboration mechanism. Macroeconomic Management. 2021;(12):63-70. doi: 10.19709/j.cnki.11-3199/f.2021.12.019.

External review comments6:(6) It is better if the authors could introduce the limitations of the study and future studies in the conclusion section. The following literature should be helpful for your research: Decoupling economic growth from water consumption in the Yangtze River Economic Belt, China. And Coordination of the Industrial-Ecological Economy in the Yangtze River Economic Belt, China.

Response6: Thanks for the kind valuable advice and the expert's advice is very reasonable. The author improved the article as a whole based on the recommended references from experts, and added the limitations of this study and future research outlook in the conclusion section. The specific changes are as follows.

This study takes 41 cities in the Yangtze River Delta as research objects, considering the agricultural development status of each city. It considers the relationship between the agricultural economy and the environment in terms of both carbon emissions and surface source pollution, which has been rare in previous studies. However, due to the limited data availability, the carbon emissions calculated in this study ignore the implied carbon emissions of agricultural production, and the index system may not fully reflect the pollution status of agricultural development. In addition, the large randomness of some original series in the gray prediction model impacts the prediction results; even if the application of the neural network model is added, it cannot accurately reflect the development status of the agricultural economy and environment in the next ten years. In future research, we will address the short-comings of this study to better understand the role of agriculture in achieving the dual carbon target.

Reviewer #2 Review Comments

External review comments1: 1- Need to replace the word "&" with "and".

Response1: Thanks for the kind valuable advice and the expert's advice is very reasonable. The author has replaced the word "&" with "and".

External review comments2: 2- The abstract is presented poorly, i suggest authors to make it more constructive.

Response2: Thanks for the kind valuable advice and the expert's advice is very reasonable. The author has modified the abstract section. The specific modifications are as follows:

Agriculture is the foundation of the national economy, and agricultural nonpoint source pollution and carbon emissions are the main environmental problems limiting the development of the agricultural economy. This study takes the Yangtze River Delta as the research object and measures agricultural carbon emissions and nonpoint source pollution in the study area from 2010 to 2020 respectively. The Tapio decoupling model is used to study types of decoupling between agricultural pollution and carbon reduction and economic growth in the Yangtze River Delta from 2010 to 2020, and the GM (1,1) model is used to predict the decoupling relationship between the agricultural environment and economic growth over the next ten years. The results show the following: (1) Agricultural COD emissions come mainly from livestock and poultry breeding, dropped from 1,130,120 tons in 2010 to 908,460 tons in 2020. Agricultural TN and TP emissions come mainly from plantations. Agricultural TN emissions dropped from 892,310 tons in 2010 to 788,020 tons in 2020. Agricultural TP emissions dropped from 149,590 tons in 2010 to130,770 tons in 2020. Agricultural carbon emissions dropped from 17,115,900 tons in 2010 to 15,786,600 tons in 2020, and come mainly from agricultural fertilizer and diesel fuel and pig breeding. (2) The decoupling effect of agricultural pollution reduction and carbon reduction in the Yangtze River Delta and economic growth has been in a long-term state, with negative decoupling occurring in a few regions, mainly in 2011, 2014 and 2020. (3) In the next ten years, except for 2021, when the coordination between agricultural pollution reduction and economic growth is poor, the two show good decoupling in the remaining years. Based on the results, this study makes recommendations on how to carry out comprehensive environmental management and promote green agricultural development.

External review comments3: 3- The introduction part need to expand little more, and clearly demonstrate the major aim of this study that what’s novel in it and how it is going to contribute in the existing literature.

Response3: Thanks for the kind valuable advice, and the expert's advice is very reasonable. The author has rewritten the introduction section. The main purpose of this study as well as the research contributions are reasonably integrated in the introduction as well as in the review section of the article, as reflected in the following.

The main purpose: The paper starts from the agricultural field and takes prefecture-level cities as the research object to explore the relationship between agricultural pollution reduction and carbon reduction and economic growth in the Yangtze River Delta region of China to provide theoretical support for realizing sustainable agricultural development in the context of "double carbon". 

Existing literature: (1) Although a more unified methodological system has been developed for the accounting of agricultural surface source pollution and agricultural carbon emissions, consistent results have not yet emerged due to different data sources and specific categories of accounting, and some results differ greatly. (2) The Central Economic Work Conference in December 2020 proposed that "we should continue to fight the battle against pollution and achieve synergistic effects of pollution reduction and carbon reduction". Studies on carbon reduction are still at the level of legal system, theoretical level and realization path, but there is a lack of corresponding empirical research and quantitative analysis. (3) The application of decoupling theory in the agricultural sector is relatively rare, especially in the study of the relationship between agricultural nonpoint source pollution and the agricultural economy. In addition, most studies take the provincial area as the basic unit or a specific city as the research object, lacking the concept of region.

Research contributions: (1) Drawing on the methods of previous studies, we measured agricultural surface source pollution and carbon emissions in the Yangtze River Delta with prefecture-level cities as the research objects, bridging the gap in previous studies. (2) The relationship between agricultural environmental pollution and economic growth from both agricultural nonpoint source pollution and carbon emissions is considered. (3) The relationship between agricultural pollution reduction and economic growth is analyzed from both temporal and spatial perspectives, providing ideas and a scientific basis for reducing three-dimensional pollution at the source and achieving synergistic management of the agricultural environment and climate. (4) The decoupling trend of agricultural pollution reduction and carbon reduction from economic growth for the next ten years is predicted, and policy suggestions are put forward, focusing on the use of agricultural technology and green finance to improve the efficiency of agricultural green production.

External review comments4: 4- In the materials and methods, need to add the url link of data sources. Further, need to improve the quality of Figures 1 and 4.

Response4: Thanks for the kind valuable advice and the expert's advice is very reasonable. The author has added the url link of data sources and has redrawn Figures 1,4 &5 as required.

Data on the gross domestic product, rural population, fertilizer application, year-end stock of livestock and poultry breeding and fish production were obtained from the 2011-2021 Shanghai Statistical Yearbook (sh.gov.cn), Jiangsu Statistical Yearbook (jiangsu.gov.cn), Zhejiang Statistical Yearbook (zj.gov.cn) and Anhui Statistical Yearbook (ah.gov.cn), as well as from cities and counties. The coefficients for pig, cattle and aquatic product production and emission were obtained from the Handbook on Accounting Methods and Coefficients for Agricultural Sources Production and Emission (2021 edition) pdf (mee.gov.cn). 

Fig 1

Fig 4

Fig 5

External review comments5: 5- The findings of the Tables 4 & 5 need more explanation by comparing it with previous published studies.

Response5: Thanks for the kind valuable advice and the expert's advice is very reasonable. The author added a comparison with the results of existing studies to the analysis of the results in Tables 4 & 5.

Table 4 shows the nonpoint source pollution emissions of various pollution sources in the Yangtze River Delta, and the trend of its emissions is similar to that found by Qiu et al. [57]. From 2010 to 2020, the COD from livestock and poultry farming in the Yangtze River Delta showed an upward and then downward trend, while COD emissions from aquaculture showed an upward trend.

TP emissions from agricultural fertilizers accounted for approximately 86.958% of agricultural TP emissions, TP emissions from agricultural fertilizers showed a decreasing trend after 2011, and TP emissions caused by livestock farming and aquaculture showed a trend of first rising and then decreasing, peaking at 16,490 and 5,330 tons in 2012 and 2016, respectively. Notably, the surface source pollution emissions calculated in this study are larger than the results of Wang et al. [58]. The main reason for the difference is the different surface source pollution emissions from cultivation.

Table 5 shows the carbon emissions of agricultural emission sources in the Yangtze River Delta from 2010 to 2020. Agricultural carbon emissions calculated in this study are lower than those found by Tian et al. [59] and closer to those found by Hu et al. [60]. The main reason is that the minimum carbon emission sources selected in each study are different; the above studies take the provincial area as the research object, while this study takes the municipal areas of the Yangtze River Delta as the research object, so there are differences in the selection of indicators. From 2010 to 2020, total agricultural carbon emissions in the Yangtze River Delta show a trend of rising and then falling. Carbon emissions from each emission source in descending order are agricultural fertilizer, agricultural irrigation, agricultural diesel, pigs, agricultural plastic film, pesticide use, cattle, sheep and farm tillage. Similar to the findings of Chen et al. [56], carbon emissions from the farming sector account for 83.320% of total carbon emissions from agriculture, much larger than those from the farming sector, of which fertilizer is the largest source.

References

[56] Chen ST, Zhang KH, Zhang YW. Measurement and decoupling effects of agricultural carbon emission performance. STATISTICS & DECISION. 2021;37(22):85-8. doi: 10.13546/j.cnki.tjyjc.2021.22.019.

[57] Qiu WW, Zhong ZB, Li ZL. Agricultural Non-point Source Pollution in China: Evaluation, Convergence Characteristics and Spatial Effects. Chin Geogr Sci. 2021;31(3):571-84. doi: 10.1007/s11769-021-1200-1. PubMed PMID: WOS:000648287300013.

[58] Wang SR, Yang DW, Sun JH, Tang LH, Wang PJ, et al. Analysis on status and characteristics of agricultural non-point source pollution in China. Water Resources Protection. 2021;37(04):140-7+72.

[59] Tian Y, Wu HT. Research on Fairness of Agricultural Carbon Emissions in China’s Major Grain Producing Areas from the Perspective of Industrial Structure. Journal of Agrotechnical Economics. 2020;(01):45-55. doi: 10.13246/j.cnki.jae.2020.01.003.

[60] Hu WL, Zhang JX, Wang HL. Characteristics and Influencing Factors of Agricultural Carbon Emission in China. STATISTICS & DECISION. 2020;36(05):56-62

External review comments6: 6- Need to expand discussion little more.

Response6: Thanks for the kind valuable advice and the expert's advice is very reasonable. The author has rewritten the discussion section based on the ideas in the full text and the advice of expert.

The results show that in the past ten years, the distribution pattern of the main sources of agricultural nonpoint source pollution in the Yangtze River Delta has changed, and the total amount of pollutant discharge has decreased. The main contribution is the reduction in agricultural COD emissions. According to Zou and Wang et al. [58, 61], agricultural COD, TN and TP come mainly from livestock and poultry breeding. Therefore, to suppress the growth of agricultural nonpoint source pollution, we should focus on reducing livestock and poultry breeding pollution. During the research period, total agricultural carbon emissions in the Yangtze River Delta were high, showing a trend of first rising and then falling, similar to the national agricultural carbon emissions according to the findings of Liu et al. [55], but the reduction was not obvious. In addition, this study found that there is a more concentrated negative decoupling between agricultural pollution and economic growth in the Yangtze River Delta in individual years. This was consistent with the findings for eastern and central China of Liu et al. [62]. They explained that it may be due to the depletion of agricultural resources and the decline in output levels. Therefore, this study makes policy recommendations in the following two aspects.

Recommendations for the green economy development of agriculture 

The integration of the three industries should be accelerated, thereby promoting the green development of agriculture. The latter cannot be achieved without a large amount of capital investment, and the government can increase support for capital policies; for example, promoting green financial development can improve resource utilization efficiency and reduce environmental pollution [63, 64]. In November 2021, the Implementation Plan for Green Financial Development in the Yangtze River Delta Ecological Green Integrated Development Demonstration Zone was officially issued, which provides great support for the green development of agriculture in the Yangtze River Delta and can further explore additional green financial measures to provide policy support for balancing economic growth and environmental protection. In addition, in terms of agricultural energy, the Yangtze River Delta region can strengthen the use of new energy sources, actively promote new energy sources such as photovoltaic and solar energy, and strive to control carbon emissions at the source to achieve the dual carbon goal [65]. With the rapid development of digital industries in the Yangtze River Delta, boosting the digital economy can effectively promote the green and low-carbon development of the city, and increasing investment in R&D can reduce agricultural pollution and improve the production environment [48, 66]. With the deep integration of one, two, and three industries, advanced digital technologies and production techniques are now widely used in agricultural production as well as pollution prevention and control; for example, artificial intelligence and re-mote sensing have played an important role in pollutant monitoring and prevention [67, 68]. Therefore, in the future, advanced technologies can be further applied to agriculture, and investments in agricultural science and technology innovation can be increased to promote rapid agricultural economic growth from the production side and to reduce environmental problems caused by agricultural development and achieve clean agricultural production from the pollution side [69, 70].

Recommendations for agricultural environmental management

Promoting the combination of the market and the government, strengthening regional environmental synergy management. First, to promote coordinated economic and environ-mental development, the Chinese government has taken a series of initiatives, including the promotion of third-party management of environmental pollution in China. As a market-based policy tool, it has broad application prospects in the Yangtze River Delta region, which has a high degree of marketization. The current degree of development of third-party treatment of environmental pollution in the Yangtze River Delta region is at a leading level nationwide [71], but in practice, problems such as inadequate market mechanisms and asymmetric market information persist, and future measures such as strengthening the monitoring of rural land quality and water environment and further stimulating market dynamics through PPP and other forms can be taken. Second, agriculture should be included in the scope of the carbon market as soon as possible by taking advantage of the relatively low cost of agricultural emission reduction [72], which can not only accelerate agricultural carbon emission reduction but also help to promote sustainable emission reduction in agricultural surface source pollution. Finally, environmental management is not the responsibility of a particular region; the relationship between mountains and water and economic interconnection requires the Yangtze River Delta to achieve comprehensive rural environmental management. Each region should not only adopt differentiated governance measures according to local conditions but also cooperate with each other to achieve synergistic governance. As the integration of the Yangtze River Delta becomes a national strategy, interregional cooperation and resource sharing are increasing. Local governments at all levels can implement advanced management techniques, promote intersectoral as well as intergroup cooperation, and give full play to both the spillover effect of technology and knowledge synergy [73] to achieve collaborative agro-environmental governance [74].

References

[55] Liu D, Zhu X, Wang Y. China's agricultural green total factor productivity based on carbon emission: An analysis of evolution trend and influencing factors. Journal of Cleaner Production. 2021;278. doi: 10.1016/j.jclepro.2020.123692.

[58] Wang SR, Yang DW, Sun JH, Tang LH, Wang PJ, et al. Analysis on status and characteristics of agricultural non-point source pollution in China. Water Resources Protection. 2021;37(04):140-7+72.

[48] Wang Q, Zhang FY. Does increasing investment in research and development promote economic growth decoupling from carbon emission growth? An empirical analysis of BRICS countries. Journal of Cleaner Production. 2020;252. doi: 10.1016/j.jclepro.2019.119853. PubMed PMID: WOS:000516777200100

[62] Liu Y, Feng C. What drives the decoupling between economic growth and energyrelated CO2 emissions in China's agricultural sector? Environmental Science and Pollution Re-search. 2021;28(32):44165-82. doi: 10.1007/s11356-021-13508-2. PubMed PMID: WOS:000639770400017.

[63] Zhang S, Wu Z, Wang Y, Hao Y. Fostering green development with green finance: An empirical study on the environmental effect of green credit policy in China. Journal of Environmental Management. 2021;296. doi: 10.1016/j.jenvman.2021.113159. PubMed PMID: WOS:000685497300005.

[64] Fang Y, Shao ZQ. Whether Green Finance Can Effectively Moderate the Green Technology Innovation Effect of Heterogeneous Environmental Regulation. International Journal of Environmental Research and Public Health. 2022;19(6). doi: 10.3390/ijerph19063646. PubMed PMID: WOS:000776864100001.

[65] Zhang Y, Yu Z, Zhang J. Spatiotemporal evolution characteristics and dynamic efficiency decomposition of carbon emission efficiency in the Yellow River Basin. Plos One. 2022;17(3). doi: 10.1371/journal.pone.0264274. PubMed PMID: WOS:000780951300016.

[66] Wang L, Chen L, Li Y. Digital economy and urban low-carbon sustainable development: the role of innovation factor mobility in China. Environmental Science and Pollution Research. 2022. doi: 10.1007/s11356-022-19182-2. PubMed PMID: WOS:000759365600010.

[67] Liu W, Xu Y, Fan D, Li Y, Shao X-F, Zheng J. Alleviating corporate environmental pollution threats toward public health and safety: The role of smart city and artificial intelligence. Safety Science. 2021;143. doi: 10.1016/j.ssci.2021.105433. PubMed PMID: WOS:000690367600023.

[68] Wang P, Yao J, Wang G, Hao F, Shrestha S, Xue B, et al. Exploring the application of artificial intelligence technology for identification of water pollution characteristics and tracing the source of water quality pollutants. Science of the Total Environment. 2019;693. doi: 10.1016/j.scitotenv.2019.07.246. PubMed PMID: WOS:000489694700047.

[69] Yi X, Lin D, Li J, Zeng J, Wang D, Yang F. Ecological treatment technology for agricultural non-point source pollution in remote rural areas of China. Environmental Science and Pollution Research. 2021;28(30):40075-87. doi: 10.1007/s11356-020-08587-6. PubMed PMID: WOS:000528989000002.

[70] Liu SJ, Xu XL. The pollution halo effect of technology spillover and pollution haven effect of economic growth in agricultural foreign trade: two sides of the same coin? Environmental Science and Pollution Research. 2021;28(16):20157-73. doi: 10.1007/s11356-020-11786-w. PubMed PMID: WOS:000605568500037.

[71] Chen HJ, Si W. Third Party Govern of Environmental Pollution in the Yangtze River Delta Region: Current Situation, Problems and Suggestion. Environmental Protection. 2020;48(20):20-3. doi: 10.14026/j.cnki.0253-9705.2020.20.004.

[72] Jin SQ, Lin Y, Niu KY. Driving Green Transformation of Agriculture with Low Carbon: Characteristics of Agricultural Carbon Emissions and Its Emission Reduction Path in China. Reform. 2021;(05):29-37.

[73] Xu L, Zhou Z, Du J. An Evolutionary Game Model for the Multi-Agent Co-Governance of Agricultural Non-Point Source Pollution Control under Intensive Management Pattern in China. International Journal of Environmental Research and Public Health. 2020;17(7). doi: 10.3390/ijerph17072472. PubMed PMID: WOS:000530763300304.

[74] Hu WJ. Improve the Yangtze River Delta regional pollution prevention and control collaboration mechanism. Macroeconomic Management. 2021;(12):63-70. doi: 10.19709/j.cnki.11-3199/f.2021.12.019.

External review comments7: 7- The Conclusions should be Conclusion and Policy recommendations by recommending some policies in the light of this study findings. Further, need to add limitations and future research directions.

Response7: Thanks for the kind valuable advice and the expert's advice is very reasonable. The author revised the article according to the experts' suggestions, in which the policy recommendations are mainly reflected in the discussion section, as described in response 6. In addition, the author added the limitations of this study and future research outlook in the conclusion section.

This study takes 41 cities in the Yangtze River Delta as research objects, considering the agricultural development status of each city. It considers the relationship between the agricultural economy and the environment in terms of both carbon emissions and surface source pollution, which has been rare in previous studies. However, due to the limited data availability, the carbon emissions calculated in this study ignore the implied carbon emissions of agricultural production, and the index system may not fully reflect the pollution status of agricultural development. In addition, the large randomness of some original series in the gray prediction model impacts the prediction results; even if the application of the neural network model is added, it cannot accurately reflect the development status of the agricultural economy and environment in the next ten years. In future research, we will address the short-comings of this study to better understand the role of agriculture in achieving the dual carbon target.

External review comments8: 8- The language of the paper should be check with the help of a native English speaker to improve its quality.

Response8: Thanks for the kind valuable advice. In order to improve the quality of the language of the article, we have polished the language of the full text through the journal's recommended touch-up agency based on expert recommendations.

August 26, 2022

Authors

---

## [Decision Letter · Decision Letter 1]

29 Nov 2022

PONE-D-22-14888R1Decoupling agriculture pollution & carbon reduction from economic growth in the Yangtze River Delta, ChinaPLOS ONE

Dear Dr. Xu,

Thank you for submitting your manuscript to PLOS ONE. After careful consideration, we feel that it has merit but does not fully meet PLOS ONE’s publication criteria as it currently stands. Therefore, we invite you to submit a revised version of the manuscript that addresses the points raised during the review process.

We look forward to receiving your revised manuscript.

Kind regards,

Nor Adilla Rashidi, Ph.D.

Academic Editor

PLOS ONE

Journal Requirements:

Reviewers' comments:

Reviewer's Responses to Questions

**Comments to the Author**

1. If the authors have adequately addressed your comments raised in a previous round of review and you feel that this manuscript is now acceptable for publication, you may indicate that here to bypass the “Comments to the Author” section, enter your conflict of interest statement in the “Confidential to Editor” section, and submit your "Accept" recommendation.

Reviewer #1: All comments have been addressed

Reviewer #2: All comments have been addressed

2. Is the manuscript technically sound, and do the data support the conclusions?

Reviewer #1: Yes

Reviewer #2: Yes

3. Has the statistical analysis been performed appropriately and rigorously? 

Reviewer #1: I Don't Know

Reviewer #2: Yes

4. Have the authors made all data underlying the findings in their manuscript fully available?

Reviewer #1: Yes

Reviewer #2: Yes

5. Is the manuscript presented in an intelligible fashion and written in standard English?

Reviewer #1: Yes

Reviewer #2: Yes

6. Review Comments to the Author

Reviewer #1: Reply 1: OK (But please briefly explain the importance of the Yangtze River Delta, and then introduce the problem of agricultural pollution in the Yangtze River Delta).

Reply 2: Please summarize the innovation points (1), (3) and (4) of this article. The workload of the paper is not innovation.

Reply 3: Please further explain the relationship between pollution reduction and carbon reduction and agricultural economic development, and use data as much as possible.

Reply 4: OK

Reply 5: OK

Reply 6: OK

Comment 1: The Result Section is only a description of the calculation results, and the Discussion Section consists of the analysis of the calculation results and policy recommendations. Please move the analysis content of the results section to the discussion section.The following literature should be helpful for your research：(1) Decoupling economic growth from water consumption in the Yangtze River Economic Belt, China.（2）Development of multidimensional water poverty in the Yangtze River Economic Belt, China

Comment 2: line451-- Please briefly describe which neural network model is selected and its calculation process.

Comment 3: line480--Please try to explain this phenomenon yourself and compare it with others'.

Comment 4: line484~531-- Policy suggestions should be based on the research results of this paper as much as possible, especially for those cities in negative decoupling.

Reviewer #2: In the current version the authors has addressed the all comments and suggestions to improve the quality of this study.

7. PLOS authors have the option to publish the peer review history of their article (what does this mean?). If published, this will include your full peer review and any attached files.

Reviewer #1: No

Reviewer #2: No

---

## [Author Response · Author response to Decision Letter 1]

4 Dec 2022

Dear Reviewers

 Thank you very much for your comments on our paper " Decoupling agriculture pollution and carbon reduction from economic growth in the Yangtze River Delta, China", your comments are very valuable to us. We have revised the paper according to the review comments, and we would like to explain each of the changes.

Reviewer #1 Review Comments

(1) Reply 1: OK (But please briefly explain the importance of the Yangtze River Delta, and then introduce the problem of agricultural pollution in the Yangtze River Delta).

Response 1: Thanks for the kind reply. The authors have explained the importance of the Yangtze River Delta in terms of national strategy and geographical location, and then we have introduced the problem of agricultural pollution in the Yangtze River Delta.

The importance of the Yangtze River Delta: (1) In terms of national strategy: The Yangtze River Delta is an important production area for commercial foodstuffs and a pilot demonstration area for sustainable agricultural development, which is strategically important in promoting national agricultural development and maintaining food security. (2) In terms of geographical location: The spread of agricultural pollutants will lead to serious soil and water pollution. The Yangtze River Delta is bordered by the Yellow Sea and the East China Sea, with a dense network of rivers and streams, and offers a geographical advantage for studying environmental pollution in agriculture.

The problem of agricultural pollution in the Yangtze River Delta: According to calculations based on data published in the 2020 China Environmental Statistics Yearbook, in 2019, the amount of agricultural fertilizer applied in the Yangtze River Delta region reached 6,642,000 tons, the number of pesticides used reached 197,010 tons, the amount of plastic film used was 297,840 tons, and the amount of agricultural COD and agricultural ammonia nitrogen in wastewater discharge accounted for 8.856% and 22.672% of the country's total, respectively, posing a serious threat to the ecology of China's watersheds.

(2) Reply 2: Please summarize the innovation points (1), (3) and (4) of this article. The workload of the paper is not innovation.

Response 2: Thanks for the kind reply. The authors have summarized the innovation points (1), (3) and (4) of this article. The specific modifications are as follows.

Given this goal, this paper makes the following four three contributions. (1) In terms of the research level, previous studies have taken the provincial-level unit or an individual city as the object of study. This paper draws on the previous research methods and takes 41 prefecture-level cities in the Yangtze River Delta as the object of study, and makes a longitudinal and cross-sectional comparison of each region, which makes up for the deficiencies of previous studies from the spatial perspective. (2) In terms of research content, previous studies have only explored the relationship between agricultural pollution and economy. This study examines the relationship between non-point source pollution from agriculture and carbon emission from the concept of pollution reduction and carbon reduction, providing ideas and scientific basis for reducing three-dimensional pollution at source and achieving synergistic management of agricultural environment and climate. (3) This study not only investigates the decoupling between agricultural pollution and economy in the Yangtze River Delta, but also predicts the decoupling trend between agricultural pollution and carbon reduction and ecological and economic growth in the next decade, which is extended in the time range.

(3) Reply 3: Please further explain the relationship between pollution reduction and carbon reduction and agricultural economic development, and use data as much as possible.

Response 3: Thanks for the kind reply. The authors have made modifications according to your comments, and the specific modifications are as follows.

(1) The development of the agricultural economy has generated agricultural pollution, and with the disappearance of the demographic dividend, environmental pollution and resource depletion have become increasingly prominent and counterproductive constraints on the development of the agricultural economy [2, 3]. Agricultural production is highly dependent on the environment and climate.

(2) Some studies have shown that COD, TN, and TP pollution loads from Chinese agriculture increased by 91.0%, 196.2%, and 244.1% from 1978 to 2017 [6], which is consistent with China's traditional agricultural model that relies on inputs of production factors to achieve yield growth. The average annual growth rate of CO2 emissions is 0.78% lower than the average annual growth rate of agricultural output of 3.82% from 1997 to 2014, which satisfies the EKC curve characteristics [7]. Incorporating agricultural carbon emissions and agricultural surface source pollution simultaneously into the three-stage SBM-DEA model, it was found that the overall AGTFP of Chinese agriculture fluctuated from 0.49 to 0.63 over time from 2000 to 2017, in which the AGTFP of all regions began to decline sharply after 2012, indicating that the conflict between agricultural production and environmental pollution in China and each region has seriously deteriorated since 2012 [8]. This study shows that the average growth rate of agricultural economy in the Yangtze River Delta decreases from 11.14% in 2011 to 5.60% in 2020, and the average growth rates of carbon, COD emissions decrease from 4.74%, 5.18% in 2011 to 1.50%, 12.44% in 2020. There is a more concentrated negative decoupling between agricultural pollution and economic growth in the YRD in individual years.

References

[2] Li S, Gong Q, Yang S. Analysis of the Agricultural Economy and Agricultural Pollution Using the Decoupling Index in Chengdu, China. International Journal of Environmental Research and Public Health. 2019;16(21). doi: 10.3390/ijerph16214233. PubMed PMID: WOS:000498842000179.

[3] Wang XB, Chen CX. The Pressure from Economic Growth Target and the Green Development Quality of Manufacturing Industry——A Measurement and Econometric Analysis Based on GTFP. Journal of Macro-quality Research. 2021;9(03):50-69. doi: 10.13948/j.cnki.hgzlyj.2021.03.005.

[6] Zou LL, Liu YS, Wang YS, Hu XD. Assessment and analysis of agricultural non-point source pollution loads in China: 1978-2017. Journal of Environmental Management. 2020;263:10. doi: 10.1016/j.jenvman.2020.110400.

[7] Luo Y, Long X, Wu C, Zhang J. Decoupling CO2 emissions from economic growth in agricultural sector across 30 Chinese provinces from 1997 to 2014. Journal of Cleaner Production. 2017;159:220-8. doi: https://doi.org/10.1016/j.jclepro.2017.05.076

[8] Chen YF, Miao JF, Zhu ZT. Measuring green total factor productivity of China's agricultural sector: A three-stage SBM-DEA model with non-point source pollution and CO2 emissions. Journal of Cleaner Production. 2021;318. doi: 10.1016/j.jclepro.2021.128543.

Comment 1: The Result Section is only a description of the calculation results, and the Discussion Section consists of the analysis of the calculation results and policy recommendations. Please move the analysis content of the results section to the discussion section. The following literature should be helpful for your research: (1) Decoupling economic growth from water consumption in the Yangtze River Economic Belt, China. (2)Development of multidimensional water poverty in the Yangtze River Economic Belt, China.

Response 1: Thanks for the kind valuable advice and the expert's advice is very reasonable. The authors have moved the analysis content of the results section to the discussion section, and the specific modifications are as follows.

Discussion

The results show that in the past ten years, the distribution pattern of the main sources of agricultural nonpoint source pollution in the YRD has changed, and the total amount of pollutant discharge has decreased. The main contribution is the reduction in agricultural COD emissions. According to Zou and Wang et al. [61, 66], agricultural COD, TN and TP come mainly from livestock and poultry breeding. Therefore, to suppress the growth of agricultural nonpoint source pollution, we should focus on reducing livestock and poultry breeding pollution. During the research period, total agricultural carbon emissions in the YRD were high, showing a trend of first rising and then falling, similar to the national agricultural carbon emissions according to the findings of Liu et al. [58], but the reduction was not obvious. In addition, this study shows that the average growth rate of agricultural economy decreases from 11.14% in 2011 to 5.60% in 2020, and the average growth rates of carbon, COD emissions decrease from 4.74%, 5.18% in 2011 to 1.50%, 12.44% in 2020, the focus of agricultural environmental management is still to reduce carbon and COD emissions in Yangtze River Delta. There is a more concentrated negative decoupling between agricultural pollution and economic growth in the YRD in individual years. This was consistent with the findings for eastern and central China of Liu et al. [67]. They explained that it may be due to the depletion of agricultural resources and the decline in output levels. We agree with the above view and believe that with the rapid development of secondary and tertiary industries, the loss of rural population and backward agricultural production has caused this phenomenon. In addition, This study concluded that the agricultural COD emissions originate mainly from livestock and aquaculture, and the livestock and aquaculture emission coefficients of the YRD cities are high in China; due mainly to the impact of swine fever in 2019 and COVID-19, the rapid increase in the number of livestock and poultry breeding in the YRD in 2020 compared with the previous year caused a substantial increase in agricultural COD emissions, but the agricultural economy has not recovered to a corresponding degree, resulting in a lack of coordination between pollution emissions and the development of the agricultural economy. Therefore, this study makes policy recommendations in the following two aspects.

Recommendations for the green economy development of agriculture 

The integration of the three industries should be accelerated, thereby promoting the green development of agriculture. The latter cannot be achieved without a large amount of capital investment, and the government can increase support for capital policies; for example, for the northern Anhui and northern Jiangsu regions, which are located in large agricultural provinces but relatively economically backward areas, promoting green financial development can not only improve resource utilization efficiency and reduce environmental pollution, but also promote local economic development [68, 69]. In November 2021, the Implementation Plan for Green Financial Development in the YRD Ecological Green Integrated Development Demonstration Zone was officially issued, which provides great support for the green development of agriculture in the YRD and can further explore additional green financial measures to provide policy support for balancing economic growth and environmental protection. In addition, in terms of agricultural energy, the YRD region can strengthen the use of new energy sources, especially in coastal areas such as Lianyungang, Yancheng and Nantong, energy consumption is high and agricultural eco-efficiency is low, should strive to control pollution emissions at the source to achieve the dual carbon goal [70]. With the rapid development of digital industries in the YRD, boosting the digital economy can effectively promote the green and low-carbon development of the city, and increasing investment in R&D can reduce agricultural pollution and improve the production environment [51, 71]. With the deep integration of one, two, and three industries, advanced digital technologies and production techniques are now widely used in agricultural production as well as pollution prevention and control; for example, artificial intelligence and remote sensing have played an important role in pollutant monitoring and prevention [72, 73]. Therefore, in the future, advanced technologies can be further applied to agriculture, and investments in agricultural science and technology innovation can be increased to promote rapid agricultural economic growth from the production side and reduce environmental problems caused by agricultural development and achieve clean agricultural production from the pollution side [74, 75].

Recommendations for agricultural environmental management

Promoting the combination of the market and the government, strengthening regional environmental synergy management. First, to promote coordinated economic and environmental development, the Chinese government has taken a series of initiatives, including the promotion of third-party management of environmental pollution in China. As a market-based policy tool, it has broad application prospects in the YRD region, which has a high degree of marketization. The current degree of development of third-party treatment of environmental pollution in the YRD region is at a leading level nationwide [76], but in practice, problems such as inadequate market mechanisms and asymmetric market information persist. Especially for cities with good urban economic development but poor agricultural environmental management, such as Shanghai and Jiaxing, future measures such as strengthening the monitoring of rural land quality and water environment and further stimulating market dynamics through PPP and other forms can be taken. Second, agriculture should be included in the scope of the carbon market as soon as possible by taking advantage of the relatively low cost of agricultural emission reduction [77], which can not only accelerate agricultural carbon emission reduction but also help to promote sustainable emission reduction in agricultural nonpoint source pollution. Finally, environmental management is not the responsibility of a particular region. It can be seen that Jiangsu and Anhui, both of which are large agricultural provinces, are in a disadvantageous position in terms of agro-ecological environment, especially in the regions bordering the two provinces, where the agricultural environment and the economy show a negative decoupling of the majority of cities. At the same time, the relationship between the mountains and water and the economy provides geographical conditions for the comprehensive management of the rural environment in many places. . Local governments at all levels can implement advanced management techniques, promote inter-sectoral as well as intergroup cooperation, and give full play to both the spillover effect of technology and knowledge synergy [78] to achieve collaborative agro-environmental governance [79].

References

[51] Wang Q, Zhang FY. Does increasing investment in research and development promote economic growth decoupling from carbon emission growth? An empirical analysis of BRICS countries. Journal of Cleaner Production. 2020;252. doi: 10.1016/j.jclepro.2019.119853. PubMed PMID: WOS:000516777200100.

[58] Liu D, Zhu X, Wang Y. China's agricultural green total factor productivity based on carbon emission: An analysis of evolution trend and influencing factors. Journal of Cleaner Production. 2021;278. doi: 10.1016/j.jclepro.2020.123692.

[61] Wang SR, Yang DW, Sun JH, Tang LH, Wang PJ, et al. Analysis on status and characteristics of agricultural non-point source pollution in China. Water Resources Protection. 2021;37(04):140-7+72.

[66] Zou L, Liu Y, Wang Y, Hu X. Assessment and analysis of agricultural non-point source pollution loads in China: 1978-2017. Journal of Environmental Management. 2020;263. doi: 10.1016/j.jenvman.2020.110400. PubMed PMID: WOS:000530232700033.

[67] Liu Y, Feng C. What drives the decoupling between economic growth and energy-related CO2 emissions in China's agricultural sector? Environmental Science and Pollution Research. 2021;28(32):44165-82. doi: 10.1007/s11356-021-13508-2. PubMed PMID: WOS:000639770400017.

[68] Zhang S, Wu Z, Wang Y, Hao Y. Fostering green development with green finance: An empirical study on the environmental effect of green credit policy in China. Journal of Environmental Management. 2021;296. doi: 10.1016/j.jenvman.2021.113159. PubMed PMID: WOS:000685497300005.

[69] Fang Y, Shao ZQ. Whether Green Finance Can Effectively Moderate the Green Technology Innovation Effect of Heterogeneous Environmental Regulation. International Journal of Environmental Research and Public Health. 2022;19(6). doi: 10.3390/ijerph19063646. PubMed PMID: WOS:000776864100001.

[70] Zhang Y, Yu Z, Zhang J. Spatiotemporal evolution characteristics and dynamic efficiency decomposition of carbon emission efficiency in the Yellow River Basin. Plos One. 2022;17(3). doi: 10.1371/journal.pone.0264274. PubMed PMID: WOS:000780951300016.

[71] Wang L, Chen L, Li Y. Digital economy and urban low-carbon sustainable development: the role of innovation factor mobility in China. Environmental Science and Pollution Research. 2022. doi: 10.1007/s11356-022-19182-2. PubMed PMID: WOS:000759365600010.

[72] Liu W, Xu Y, Fan D, Li Y, Shao X-F, Zheng J. Alleviating corporate environmental pollution threats toward public health and safety: The role of smart city and artificial intelligence. Safety Science. 2021;143. doi: 10.1016/j.ssci.2021.105433. PubMed PMID: WOS:000690367600023.

[73] Wang P, Yao J, Wang G, Hao F, Shrestha S, Xue B, et al. Exploring the application of artificial intelligence technology for identification of water pollution characteristics and tracing the source of water quality pollutants. Science of the Total Environment. 2019;693. doi: 10.1016/j.scitotenv.2019.07.246. PubMed PMID: WOS:000489694700047.

[74] Yi X, Lin D, Li J, Zeng J, Wang D, Yang F. Ecological treatment technology for agricultural non-point source pollution in remote rural areas of China. Environmental Science and Pollution Research. 2021;28(30):40075-87. doi: 10.1007/s11356-020-08587-6. PubMed PMID: WOS:000528989000002.

[75] Liu SJ, Xu XL. The pollution halo effect of technology spillover and pollution haven effect of economic growth in agricultural foreign trade: two sides of the same coin? Environmental Science and Pollution Research. 2021;28(16):20157-73. doi: 10.1007/s11356-020-11786-w. PubMed PMID: WOS:000605568500037.

[76] Chen HJ, Si W. Third Party Govern of Environmental Pollution in the Yangtze River Delta Region: Current Situation, Problems and Suggestion. Environmental Protection. 2020;48(20):20-3. doi: 10.14026/j.cnki.0253-9705.2020.20.004.

[77] Jin SQ, Lin Y, Niu KY. Driving Green Transformation of Agriculture with Low Carbon: Characteristics of Agricultural Carbon Emissions and Its Emission Reduction Path in China. Reform. 2021;(05):29-37.

[78] Xu L, Zhou Z, Du J. An Evolutionary Game Model for the Multi-Agent Co-Governance of Agricultural Non-Point Source Pollution Control under Intensive Management Pattern in China. International Journal of Environmental Research and Public Health. 2020;17(7). doi: 10.3390/ijerph17072472. PubMed PMID: WOS:000530763300304.

[79] Hu WJ. Improve the Yangtze River Delta regional pollution prevention and control collaboration mechanism. Macroeconomic Management. 2021;(12):63-70. doi: 10.19709/j.cnki.11-3199/f.2021.12.019.

Comment 2: line451-- Please briefly describe which neural network model is selected and its calculation process.

Response 2: Thanks for the kind valuable advice and the expert's advice is very reasonable.

In order to ensure a good prediction effect, this paper mainly combines the BP neural network model to train the prediction results with unqualified accuracy, However, this model is used in this study as an auxiliary tool to improve the auxiliary effect of the gray correlation model, and is not the main method of the article. The formulas are not listed one by one in the Methodology section to ensure the length is within a certain range. The specific method comes from references[64,65].

References

[64] Hu Z, Gong X, Liu H. Prediction of household consumption carbon emission in western cities Based on BP model: Case of Xi'an city. Journal of Arid Land Resources and Environment. 2020;34(07):82-9. doi: 10.13448/j.cnki.jalre.2020.187.

[65] Yuan L, Yang D, Wu X, He W, Kong Y, Ramsey TS, et al. Development of multidimensional water poverty in the Yangtze River Economic Belt, China. Journal of Environmental Management. 2023;325:116608. doi: https://doi.org/10.1016/j.jenvman.2022.116608.

Comment 3: line480--Please try to explain this phenomenon yourself and compare it with others'.

Response 3: Thanks for the kind valuable advice and the expert's advice is very reasonable. The authors have made modifications according to your comments, and the specific modifications are as follows.

Discussion

The results show that in the past ten years, the distribution pattern of the main sources of agricultural nonpoint source pollution in the YRD has changed, and the total amount of pollutant discharge has decreased. The main contribution is the reduction in agricultural COD emissions. According to Zou and Wang et al. [61, 66], agricultural COD, TN and TP come mainly from livestock and poultry breeding. Therefore, to suppress the growth of agricultural nonpoint source pollution, we should focus on reducing livestock and poultry breeding pollution. During the research period, total agricultural carbon emissions in the YRD were high, showing a trend of first rising and then falling, similar to the national agricultural carbon emissions according to the findings of Liu et al. [58], but the reduction was not obvious. In addition, this study shows that the average growth rate of agricultural economy decreases from 11.14% in 2011 to 5.60% in 2020, and the average growth rates of carbon, COD emissions decrease from 4.74%, 5.18% in 2011 to 1.50%, 12.44% in 2020, the focus of agricultural environmental management is still to reduce carbon and COD emissions in Yangtze River Delta. There is a more concentrated negative decoupling between agricultural pollution and economic growth in the YRD in individual years. This was consistent with the findings for eastern and central China of Liu et al. [67]. They explained that it may be due to the depletion of agricultural resources and the decline in output levels. We agree with the above view and believe that with the rapid development of secondary and tertiary industries, the loss of rural population and backward agricultural production has caused this phenomenon. In addition, This study concluded that the agricultural COD emissions originate mainly from livestock and aquaculture, and the livestock and aquaculture emission coefficients of the YRD cities are high in China; due mainly to the impact of swine fever in 2019 and COVID-19, the rapid increase in the number of livestock and poultry breeding in the YRD in 2020 compared with the previous year caused a substantial increase in agricultural COD emissions, but the agricultural economy has not recovered to a corresponding degree, resulting in a lack of coordination between pollution emissions and the development of the agricultural economy. Therefore, this study makes policy recommendations in the following two aspects.

References

[58] Liu D, Zhu X, Wang Y. China's agricultural green total factor productivity based on carbon emission: An analysis of evolution trend and influencing factors. Journal of Cleaner Production. 2021;278. doi: 10.1016/j.jclepro.2020.123692.

[61] Wang SR, Yang DW, Sun JH, Tang LH, Wang PJ, et al. Analysis on status and characteristics of agricultural non-point source pollution in China. Water Resources Protection. 2021;37(04):140-7+72.

[66] Zou L, Liu Y, Wang Y, Hu X. Assessment and analysis of agricultural non-point source pollution loads in China: 1978-2017. Journal of Environmental Management. 2020;263. doi: 10.1016/j.jenvman.2020.110400. PubMed PMID: WOS:000530232700033.

[67] Liu Y, Feng C. What drives the decoupling between economic growth and energy-related CO2 emissions in China's agricultural sector? Environmental Science and Pollution Research. 2021;28(32):44165-82. doi: 10.1007/s11356-021-13508-2. PubMed PMID: WOS:000639770400017.

Comment 4: line484~531-- Policy suggestions should be based on the research results of this paper as much as possible, especially for those cities in negative decoupling.

Response 4: Thanks for the kind valuable advice and the expert's advice is very reasonable.

Recommendations for the green economy development of agriculture 

The integration of the three industries should be accelerated, thereby promoting the green development of agriculture. The latter cannot be achieved without a large amount of capital investment, and the government can increase support for capital policies; for example, for the northern Anhui and northern Jiangsu regions, which are located in large agricultural provinces but relatively economically backward areas, promoting green financial development can not only improve resource utilization efficiency and reduce environmental pollution, but also promote local economic development [68, 69]. In November 2021, the Implementation Plan for Green Financial Development in the YRD Ecological Green Integrated Development Demonstration Zone was officially issued, which provides great support for the green development of agriculture in the YRD and can further explore additional green financial measures to provide policy support for balancing economic growth and environmental protection. In addition, in terms of agricultural energy, the YRD region can strengthen the use of new energy sources, especially in coastal areas such as Lianyungang, Yancheng and Nantong, energy consumption is high and agricultural eco-efficiency is low, should strive to control pollution emissions at the source to achieve the dual carbon goal [70]. With the rapid development of digital industries in the YRD, boosting the digital economy can effectively promote the green and low-carbon development of the city, and increasing investment in R&D can reduce agricultural pollution and improve the production environment [51, 71]. With the deep integration of one, two, and three industries, advanced digital technologies and production techniques are now widely used in agricultural production as well as pollution prevention and control; for example, artificial intelligence and remote sensing have played an important role in pollutant monitoring and prevention [72, 73]. Therefore, in the future, advanced technologies can be further applied to agriculture, and investments in agricultural science and technology innovation can be increased to promote rapid agricultural economic growth from the production side and reduce environmental problems caused by agricultural development and achieve clean agricultural production from the pollution side [74, 75].

Recommendations for agricultural environmental management

Promoting the combination of the market and the government, strengthening regional environmental synergy management. First, to promote coordinated economic and environmental development, the Chinese government has taken a series of initiatives, including the promotion of third-party management of environmental pollution in China. As a market-based policy tool, it has broad application prospects in the YRD region, which has a high degree of marketization. The current degree of development of third-party treatment of environmental pollution in the YRD region is at a leading level nationwide [76], but in practice, problems such as inadequate market mechanisms and asymmetric market information persist. Especially for cities with good urban economic development but poor agricultural environmental management, such as Shanghai and Jiaxing, future measures such as strengthening the monitoring of rural land quality and water environment and further stimulating market dynamics through PPP and other forms can be taken. Second, agriculture should be included in the scope of the carbon market as soon as possible by taking advantage of the relatively low cost of agricultural emission reduction [77], which can not only accelerate agricultural carbon emission reduction but also help to promote sustainable emission reduction in agricultural nonpoint source pollution. Finally, environmental management is not the responsibility of a particular region. It can be seen that Jiangsu and Anhui, both of which are large agricultural provinces, are in a disadvantageous position in terms of agro-ecological environment, especially in the regions bordering the two provinces, where the agricultural environment and the economy show a negative decoupling of the majority of cities. At the same time, the relationship between the mountains and water and the economy provides geographical conditions for the comprehensive management of the rural environment in many places. . Local governments at all levels can implement advanced management techniques, promote inter-sectoral as well as intergroup cooperation, and give full play to both the spillover effect of technology and knowledge synergy [78] to achieve collaborative agro-environmental governance [79].

References

[51] Wang Q, Zhang FY. Does increasing investment in research and development promote economic growth decoupling from carbon emission growth? An empirical analysis of BRICS countries. Journal of Cleaner Production. 2020;252. doi: 10.1016/j.jclepro.2019.119853. PubMed PMID: WOS:000516777200100.

[68] Zhang S, Wu Z, Wang Y, Hao Y. Fostering green development with green finance: An empirical study on the environmental effect of green credit policy in China. Journal of Environmental Management. 2021;296. doi: 10.1016/j.jenvman.2021.113159. PubMed PMID: WOS:000685497300005.

[69] Fang Y, Shao ZQ. Whether Green Finance Can Effectively Moderate the Green Technology Innovation Effect of Heterogeneous Environmental Regulation. International Journal of Environmental Research and Public Health. 2022;19(6). doi: 10.3390/ijerph19063646. PubMed PMID: WOS:000776864100001.

[70] Zhang Y, Yu Z, Zhang J. Spatiotemporal evolution characteristics and dynamic efficiency decomposition of carbon emission efficiency in the Yellow River Basin. Plos One. 2022;17(3). doi: 10.1371/journal.pone.0264274. PubMed PMID: WOS:000780951300016.

[71] Wang L, Chen L, Li Y. Digital economy and urban low-carbon sustainable development: the role of innovation factor mobility in China. Environmental Science and Pollution Research. 2022. doi: 10.1007/s11356-022-19182-2. PubMed PMID: WOS:000759365600010.

[72] Liu W, Xu Y, Fan D, Li Y, Shao X-F, Zheng J. Alleviating corporate environmental pollution threats toward public health and safety: The role of smart city and artificial intelligence. Safety Science. 2021;143. doi: 10.1016/j.ssci.2021.105433. PubMed PMID: WOS:000690367600023.

[73] Wang P, Yao J, Wang G, Hao F, Shrestha S, Xue B, et al. Exploring the application of artificial intelligence technology for identification of water pollution characteristics and tracing the source of water quality pollutants. Science of the Total Environment. 2019;693. doi: 10.1016/j.scitotenv.2019.07.246. PubMed PMID: WOS:000489694700047.

[74] Yi X, Lin D, Li J, Zeng J, Wang D, Yang F. Ecological treatment technology for agricultural non-point source pollution in remote rural areas of China. Environmental Science and Pollution Research. 2021;28(30):40075-87. doi: 10.1007/s11356-020-08587-6. PubMed PMID: WOS:000528989000002.

[75] Liu SJ, Xu XL. The pollution halo effect of technology spillover and pollution haven effect of economic growth in agricultural foreign trade: two sides of the same coin? Environmental Science and Pollution Research. 2021;28(16):20157-73. doi: 10.1007/s11356-020-11786-w. PubMed PMID: WOS:000605568500037.

[76] Chen HJ, Si W. Third Party Govern of Environmental Pollution in the Yangtze River Delta Region: Current Situation, Problems and Suggestion. Environmental Protection. 2020;48(20):20-3. doi: 10.14026/j.cnki.0253-9705.2020.20.004.

[77] Jin SQ, Lin Y, Niu KY. Driving Green Transformation of Agriculture with Low Carbon: Characteristics of Agricultural Carbon Emissions and Its Emission Reduction Path in China. Reform. 2021;(05):29-37.

[78] Xu L, Zhou Z, Du J. An Evolutionary Game Model for the Multi-Agent Co-Governance of Agricultural Non-Point Source Pollution Control under Intensive Management Pattern in China. International Journal of Environmental Research and Public Health. 2020;17(7). doi: 10.3390/ijerph17072472. PubMed PMID: WOS:000530763300304.

[79] Hu WJ. Improve the Yangtze River Delta regional pollution prevention and control collaboration mechanism. Macroeconomic Management. 2021;(12):63-70. doi: 10.19709/j.cnki.11-3199/f.2021.12.019.

Reviewer #2 Review Comments

Reviewer #2: In the current version the authors have addressed all comments and suggestions to improve the quality of this study.

Response: Thank you very much for taking the time to review our article and give your valuable comments, which are of great importance to this study. We appreciate your approval.

---

## [Decision Letter · Decision Letter 2]

26 Dec 2022

Decoupling agriculture pollution & carbon reduction from economic growth in the Yangtze River Delta, China

PONE-D-22-14888R2

Dear Dr. Xu,

We’re pleased to inform you that your manuscript has been judged scientifically suitable for publication and will be formally accepted for publication once it meets all outstanding technical requirements.

Kind regards,

Nor Adilla Rashidi, Ph.D.

Academic Editor

PLOS ONE

Additional Editor Comments (optional):

Reviewers' comments:

Reviewer's Responses to Questions

**Comments to the Author**

1. If the authors have adequately addressed your comments raised in a previous round of review and you feel that this manuscript is now acceptable for publication, you may indicate that here to bypass the “Comments to the Author” section, enter your conflict of interest statement in the “Confidential to Editor” section, and submit your "Accept" recommendation.

Reviewer #2: All comments have been addressed

2. Is the manuscript technically sound, and do the data support the conclusions?

Reviewer #2: Yes

3. Has the statistical analysis been performed appropriately and rigorously? 

Reviewer #2: Yes

4. Have the authors made all data underlying the findings in their manuscript fully available?

Reviewer #2: Yes

5. Is the manuscript presented in an intelligible fashion and written in standard English?

Reviewer #2: Yes

6. Review Comments to the Author

Reviewer #2: Current version is good, because the authors have been improved the quality of the paper by following my suggestions and comments.

7. PLOS authors have the option to publish the peer review history of their article (what does this mean?). If published, this will include your full peer review and any attached files.

Reviewer #2: No

---

## [Editor Report · Acceptance letter]

6 Jan 2023

PONE-D-22-14888R2 

Decoupling agriculture pollution and carbon reduction from economic growth in the Yangtze River Delta, China 

Dear Dr. Xu:

I'm pleased to inform you that your manuscript has been deemed suitable for publication in PLOS ONE. Congratulations! Your manuscript is now with our production department. 

Kind regards, 

on behalf of

Dr. Nor Adilla Rashidi 

Academic Editor

PLOS ONE